# The Hidden Risk: Membership Inference Attacks on Multimodal Federated Learning via Modality Imbalance

**Chang Ma** [1]  **Jun Li** [2]  **Kang Wei** [3]  **Yipeng Zhou** [4]  **Ming Ding** [5]  **Yiyang Ni** [6][7]

## Abstract

Federated learning (FL) faces significant challenges from modality heterogeneity, which has motivated multimodal federated learning (MFL) to leverage complementary modalities across decentralized clients for improved performance. However, modality imbalance introduces a new attack surface, making MFL more vulnerable to membership inference attacks (MIAs), an issue that remains largely unexplored. In this work, we present the first systematic study of MIAs against MFL and propose a modality-aware attack framework. We show that multimodal models are inherently more susceptible to MIAs due to heterogeneous modality contributions, and existing attacks are suboptimal as they treat multimodal parameters as a whole. By performing MIAs on individual modalities, we find that (i) attacking the dominant modality achieves comparable accuracy with lower overhead, and (ii) different modalities expose distinct membership patterns. To identify members with different patterns, we propose a modality-aware framework that exploits cross-modal performance gaps to adaptively select attack modalities and calibrate inference results. Experiments on three datasets show our approach outperforms baselines across multiple metrics.

## 1. Introduction

Federated learning (FL) is a collaborative training paradigm that enables multiple clients to jointly learn a model without sharing raw data, and has been widely adopted in privacy-sensitive domains such as healthcare and finance (Yang et al., 2019; Wei et al., 2021; 2023). Recently, multimodal FL (MFL) has attracted growing attention for exploiting heterogeneous modality information to achieve superior performance (Qi & Li, 2024; Ma et al., 2025). Unlike unimodal FL where clients exchange a single model's parameters, each client in MFL maintains multiple modality-specific encoders and a classifier. Each encoder extracts modality-specific features, which are fused and passed to the classifier for prediction. Despite keeping raw data local, MFL remains vulnerable to membership inference attacks (MIAs), as the exchange of gradients and model parameters between clients and the server exposes additional attack surfaces compared to centralized learning (Bai et al., 2024). MIAs (Nasr et al., 2018) pose a serious privacy threat by inferring whether a specific data sample was included in the training set. For example, an attacker may infer whether an individual has a particular disease by determining whether their medical records were used to train a disease prediction model.

Existing MIAs typically treat the exchanged parameters as a whole and infer membership based on metrics such as gradient norms or per-sample loss values computed using these parameters, without disrupting model updates (Yeom et al., 2018; Li et al., 2023; Zhu et al., 2025). Recent advances further improve attack accuracy by incorporating difficulty calibration, which contrasts each sample's behavior on the target client with that on other clients and leverages the resulting discrepancies to refine membership inference (Sablayrolles et al., 2019; Shi et al., 2024; He et al., 2024). Existing MIAs overlook the multimodal setting, where parameters from different encoders exhibit heterogeneous optimization speeds and performance, a phenomenon known as modality imbalance. Treating multimodal parameters as a monolithic entity overlooks modality-specific information, raising the question of whether such heterogeneity further increases the vulnerability of MFL to MIAs and how it can be effectively exploited to strengthen attacks.

To address this gap, we systematically investigate the vulner-

---

[1]School of Cyber Science and Engineering, Nanjing University of Science and Technology, Nanjing, China [2]School of Information Science and Engineering, Southeast University, Nanjing, China [3]School of Cyber Science and Engineering, Southeast University, Nanjing, China [4]School of Computing, Macquarie University, Sydney, Australia [5]CSIRO, Sydney, NSW, Australia [6]Institute of Artificial Intelligence Research, Jiangsu Second Normal University, Nanjing, China [7]Jiangsu Key Laboratory of Wireless Communications, Nanjing University of Posts and Telecommunications, Nanjing, China. Correspondence to: Yiyang Ni <niyy@jssnu.edu.cn>, Kang Wei <kang.wei@seu.edu.cn>.

*Proceedings of the $43^{rd}$ International Conference on Machine Learning*, Seoul, South Korea. PMLR 306, 2026. Copyright 2026 by the author(s).

ability of MFL to MIAs and propose a novel MIA method tailored to multimodal settings. Our theoretical analysis and empirical study reveals that: (1) due to optimization imbalance across heterogeneous modalities, a dominant modality converges faster than others and tends to overfit as training progresses, making multimodal models inherently more vulnerable to MIAs; (2) when existing MIAs are applied to MFL, they fail to fully exploit modality-specific information; (3) comparable or even higher attack accuracy can be achieved with significantly lower computational overhead by attacking only the dominant modality, as it largely dominates the model's behavior; (4) non-dominant modalities remain valuable, as they enable the identification of a minority of member samples that exhibit preferences toward non-dominant modalities, i.e., samples whose quality is higher in these modalities.

Based on these findings, we propose a novel MIA method that adaptively weights and calibrates MIA scores, i.e., membership likelihood values produced by different attack strategies, for samples with diverse modality preferences. Specifically, we construct an attack model that evaluates each sample's cross-modal performance discrepancy and assigns adaptive weights to modality-specific MIA scores accordingly. The model then aggregates the weighted scores and calibrates them using these discrepancies to generate final membership predictions. The main contributions of this work are summarized as follows:

- We demonstrate that optimization imbalance across modalities significantly increases the vulnerability of MFL to MIAs, yet existing MIAs fail to fully exploit modality-specific information.

- We propose a novel MIA framework that adaptively selects and calibrates attack scores for each sample based on its cross-modal performance discrepancies.

- Extensive experiments on three real-world datasets show that our method consistently improves MIA accuracy across samples with diverse modality preferences.

## 2. Related Work

### 2.1. Imbalanced Multimodal Federated Learning

MFL leverages information from different modalities to achieve superior performance, while it introduces the phenomenon of modality imbalance, where one dominant modality tends to optimize rapidly during the learning process and suppress the optimization of other modalities (Xia et al., 2023; Zhang et al., 2023b). Existing work has explored solutions to the challenges posed by modality imbalance. Fan et al. (2024) addressed the performance degradation in MFL caused by modality imbalance by adjusting the optimization speed of different modalities. Ma et al. (2025)

mitigated the more severe performance degradation from modality imbalance when applying differential privacy (DP) to MFL. Shang et al. (2023) alleviated the increased vulnerability to adversarial attacks in multimodal models caused by modality imbalance. However, the privacy implications of modality imbalance in MFL remain underexplored.

### 2.2. Membership Inference Attack

MIAs infer whether a specific sample was used to train a target model (Shokri et al., 2017). These attacks exploit behavioral differences between training and non-training data by computing membership scores from model parameters such as gradients and loss values (Yeom et al., 2018; Li et al., 2023). Recent advances incorporate difficulty calibration (Sablayrolles et al., 2019; Watson et al., 2021), which improves attack accuracy at low false positive rates by contrasting sample-specific behavior against reference distributions. In FL, the exchange of model updates between clients and the server creates new attack surfaces for MIAs (Nasr et al., 2019). Li et al. (2023) exploited gradient orthogonality in exchanged model updates to identify client membership. Zhu et al. (2025) further enhanced attack effectiveness by leveraging models from non-target clients. Despite these advances in unimodal FL, MIAs targeting MFL remain largely unexplored. In this work, we bridge this gap by systematically investigating how the heterogeneous parameters exchanged across different modalities can be exploited to enhance MIA performance in MFL.

## 3. Multimodal MIA Analysis

In this section, we first investigate the vulnerability of multimodal models with imbalanced modalities, and then empirically examine the limitations of existing MIAs in the MFL setting. Due to space constraints, all proofs in this section are provided in Appendix A

### 3.1. Cause of MIA Vulnerability: Modality Imbalance

MIAs exploit the overfitting tendencies of machine learning models by leveraging behavioral discrepancies between training and non-training samples to infer membership status. To analyze the vulnerability of multimodal models under modality imbalance , we theoretically analyze their overfitting behavior by examining the generalization bound and the empirical error it contains.

Consider a supervised learning task with training dataset $D_{\text{train}} = \{(\mathbf{x}_i, y_i)\}_{i=1}^N$, where $\mathbf{x}_i \in \mathcal{X}$ are inputs, $y_i \in \mathcal{Y}$ are labels, and $N$ is the sample size. Let $f : \mathcal{X} \to \mathcal{Y}$ be a model in hypothesis class $\mathcal{H} = \{h : \mathcal{X} \to \{-1, +1\}\}$. Overfitting is quantified by the generalization and empirical

error, which is bounded with probability at least $1 - \delta$ by:

$$L(f(\mathbf{x}), y) \le \hat{L}(f(\mathbf{x}), y) + 2\mathfrak{R}_N(\mathcal{F}) + \sqrt{\frac{\log(1/\delta)}{2N}}, \quad (1)$$

where $L(f(\mathbf{x}))$ and $\hat{L}(f(\mathbf{x}))$ are the generalization and empirical errors, respectively, and $\mathfrak{R}_N(\mathcal{F})$ represents model complexity. When overfitting occurs, the empirical error $\hat{L}(f(\mathbf{x}))$ approaches zero, but model over-parameterization or insufficient training samples prevent the generalization error from decreasing correspondingly. We extend this analysis to the multimodal setting, where each input sample $\mathbf{x}_i = \{\mathbf{x}_i^m\}_{m=1}^{|M|}$ comprises $|M|$ distinct modalities. We consider a late fusion architecture where the model integrates decision-level outputs from individual modality encoders. Formally, the joint logit output is defined as:

$$f(\mathbf{x}) = \sum_{m=1}^{M} w^m f^m(\mathbf{x}^m), \quad (2)$$

where $f^m$ is the encoder for modality $m$ and $w^m$ is its fusion weight. In the following theorem, we present the generalization bound for the multimodal model.

**Theorem 3.1.** *For convex loss $L(\cdot, \cdot)$, the generalization error satisfies with probability at least $1 - \delta$:*

$$L(f(\mathbf{x}), y) \le \underbrace{\sum_{m=1}^{M} \mathbb{E}[w^m] \mathbb{E}\left[\hat{L}(f^m)\right]}_{\textit{Empirical Error}} + \sum_{m=1}^{M} \mathbb{E}[w^m] \mathfrak{R}_m(\mathcal{F})$$

$$+ \underbrace{\sum_{m=1}^{M} \mathrm{Cov}(w^m, L(f^m))}_{\textit{Addition Covariance Term}} + M\sqrt{\frac{\log(1/\delta)}{2N}},$$

$$(3)$$

*where $L(f^m)$ and $\hat{L}(f^m)$ are shorthand for $L(f^m(\mathbf{x}^m), y)$ and $\hat{L}(f^m(\mathbf{x}^m), y)$, respectively, and $\mathrm{Cov}(w^m, L(f^m))$ denotes the covariance between the fusion weight and the loss of the corresponding modality.*

We observe that the generalization bound includes an additional covariance term, which captures how modality imbalance induces correlation between fusion weights and modality-specific losses. Specifically, modalities with lower losses $L(f^m)$ receive larger fusion weights $w^m$, resulting in a negative covariance. We attribute this term to the optimization strategy adopted, meaning all components on the right-hand side except the empirical error are fixed prior to training. Consequently, we next analyze the convergence of empirical errors for modalities with different fusion weights.

**Theorem 3.2.** *Let $\hat{L}(\boldsymbol{\theta}_T^m)$ denote the empirical error of modality $m$ after $T$ training iterations, where $\boldsymbol{\theta}_T^m$ represents the model parameters of modality $m$ at this point. Under the L-smoothness assumption (detailed in Appendix A.2),*

*we have:*

$$\mathbb{E}\left[\hat{L}(\boldsymbol{\theta}_T^m)\right] \le \mathbb{E}\left[\hat{L}(\boldsymbol{\theta}_0^m)\right] - \frac{\eta w^m}{2}(1 - 2L\eta w^m)$$

$$\times \sum_{t=1}^{T} \mathbb{E}\left[\|\nabla\hat{L}(\boldsymbol{\theta}_{t-1}^m)\|^2\right] + \frac{T\eta w^m}{2}(1 + 2L\eta w^m)\lambda^m,$$

$$(4)$$

*where $\eta$ is the learning rate, and $\lambda^m$ captures the gradient difference between multimodal and unimodal training for modality $m$.*

Our analysis indicates that modalities with higher weights benefit from accelerated optimization and superior performance, which can be explained as follows. In the setting $\eta \le \frac{1}{4L}$, the empirical loss decreases at a rate proportional to the fusion weight $w^m$. Moreover, because multimodal output inherently favors modalities with larger $w^m$, these modalities exhibit smaller gradient deviation from unimodal training, i.e., smaller $\lambda^m$.

**Insight 1: Vulnerability Due to Imbalanced Modalities.** Combining the results of Theorems 3.1 and 3.2, we observe that dominant modalities exhibit faster convergence rates, lower empirical errors, and larger weights in the generalization bound. As empirical error rapidly approaches zero while the generalization error remains constrained by fixed terms that increase with additional modalities (e.g., $M\sqrt{\frac{\log(1/\delta)}{2N}}$), the multimodal model develops a large generalization gap, indicating severe overfitting. More critically, this overfitting is unavoidable via early stopping. Due to asynchronous convergence, early stopping sacrifices performance of unconverged weaker modalities, while continued training exacerbates overfitting in dominant ones. This makes multimodal models inherently vulnerable to MIAs.

**Insight 2: Gaps of Existing Work.** Our theoretical analysis reveals that different modalities exhibit distinct overfitting characteristics, indicating that each modality contains different membership information. In MFL, where all model parameters are available, this leads to greater privacy leakage. However, we observe that existing MIA methods rely on parameters from the entire model to compute attack scores (e.g., gradient updates or model outputs), while the variations in these parameters are predominantly driven by the dominant modality. Consequently, existing methods fail to adequately exploit information from all modalities, which motivates us to rethink membership inference from a modality-aware perspective.

### 3.2. Rethinking MIA Methods in Multimodal FL

Table 1[1] compares the performance disparities of several existing MIA methods, including Cosine-LIRA and Loss-LIRA (Zhu et al., 2025), Grad-Cosine (Li et al., 2023), and

---

[1]Results in Tables 1 and 2 are derived from models trained on the CREMA-D dataset (see Section 5 for details).

*Table 1.* AUC scores of MIA methods against different target modalities (higher values indicate better overall accuracy).

| Attack Method | Multimodal | Audio | Visual |
|---|---|---|---|
| Cosine-LIRA | 0.7500 | 0.7551 | 0.5140 |
| Grad-Cosine | 0.7332 | 0.7366 | 0.5185 |
| Loss-LIRA | 0.6989 | 0.6787 | 0.5126 |
| Loss-Based | 0.6629 | 0.6181 | 0.5003 |
| *Random Guess* | 0.5000 | 0.5000 | 0.5000 |

*Table 2.* MIA accuracy rate (%) when using the 0.95 quantile as decision threshold for samples with different modality preferences (Audio-Pref: Audio-preferred, Visual-Pref: Visual-preferred, Mem.: Member, Non-m.: Non-member).

| Target Modality | Preference | Loss-Based | | Loss-LiRA | |
|---|---|---|---|---|---|
| | | Mem. | Non-m. | Mem. | Non-m. |
| Audio Modality | Audio-pref. | 13.7 | 96.6 | 7.1 | 94.5 |
| | Visual-pref. | 0.0 | 98.2 | 0.0 | 100.0 |
| Visual Modality | Audio-pref. | 5.0 | 95.5 | 2.7 | 97.3 |
| | Visual-pref. | 20.0 | 93.0 | 40.0 | 82.0 |
| Multimodal | Audio-pref. | 13.4 | 96.5 | 7.4 | 94.5 |
| | Visual-pref. | 0.0 | 98.2 | 0.0 | 100.0 |

Loss-Based (Yeom et al., 2018), when targeting parameters associated with different modalities (detailed descriptions of these methods are available in Appendix B.4).

**Insight 3: Effectiveness of Dominant Modality** Our results demonstrate that conducting MIAs solely on the dominant modality achieves performance comparable to multimodal attacks while significantly reducing computational overhead (e.g., single forward pass vs. multi-channel). Moreover, focusing on the dominant modality alone even yields superior results for certain MIA methods. This phenomenon can be attributed to interference from non-dominant modalities. Specifically, these modalities introduce redundant dimensions that merely expand the feature space without offering discriminative utility (e.g., in cosine similarity calculations), thereby hindering rather than enhancing the inference.

**Insight 4: The Role of Non-dominanat Modalities.** While targeting the non-dominant modality yields inferior overall performance, we find that this approach still holds value for specific samples. Based on the disparity in modality preferences, we categorize samples into distinct preference groups according to their performance across different modalities. As illustrated in Table 2, while MIAs targeting the dominant modality achieve superior performance on samples with dominant-modality preference, they tend to trivially misclassify non-dominant-preferred samples as non-members. Conversely, attacks targeting the non-dominant modality

demonstrate a certain capability in identifying samples with non-dominant modality preference, thereby providing novel perspectives for membership inference. This observation motivates us to develop an adaptive MIA method that selects appropriate attack scores based on each sample's modality performance characteristics, as we detail in the next section.

## 4. Attack Methodology

In this section, we first define the threat model under which our attack operates. Next, we introduce the design intuition, which is based on the observation that samples with different modality preferences exhibit distinct behaviors in MIA scores. Finally, we present the overall framework and technical details of our proposed MIA method.

### 4.1. Threat Model

We consider a horizontal FL environment (Wei et al., 2020; Zhu et al., 2025) where model aggregation is performed via FedAvg (McMahan et al., 2017). We assume the existence of colluding participants who act as passive but curious observers. These adversaries intercept the model updates transmitted between clients and the server to extract privacy-sensitive information. The objective of these adversaries is to determine whether a given sample was included in the private training set of a specific target client, where the input samples are collected from a mixture of both FL-participating and non-participating data.

### 4.2. Design Intuition

The performance disparity of MIA methods on samples with different modality preferences arises from the predominance of dominant-modality-preferred samples in the dataset. To optimize overall performance, multimodal models disproportionately learn from dominant-preferred samples, making them more easily identifiable. Drawing inspiration from difficulty calibration, which calibrates scores based on samples' learning difficulty, we propose to differentiate samples based on their modality preferences and apply modality-aware membership inference.

Figure 1 demonstrates the intuition behind our approach. First, we observe that categorizing samples according to their modality preferences enhances the discriminability of membership scores that were originally ambiguous. For visual-preferred samples, we find that multiple attack methods can distinguish between members and non-members, yet these discriminative signals are often underutilized in existing approaches. Specifically, when using loss-Based MIA on the audio modality, although samples of all modality preferences demonstrate reasonable discriminability, the membership decision threshold for visual-preferred samples is substantially lower than that for audio-preferred sam-

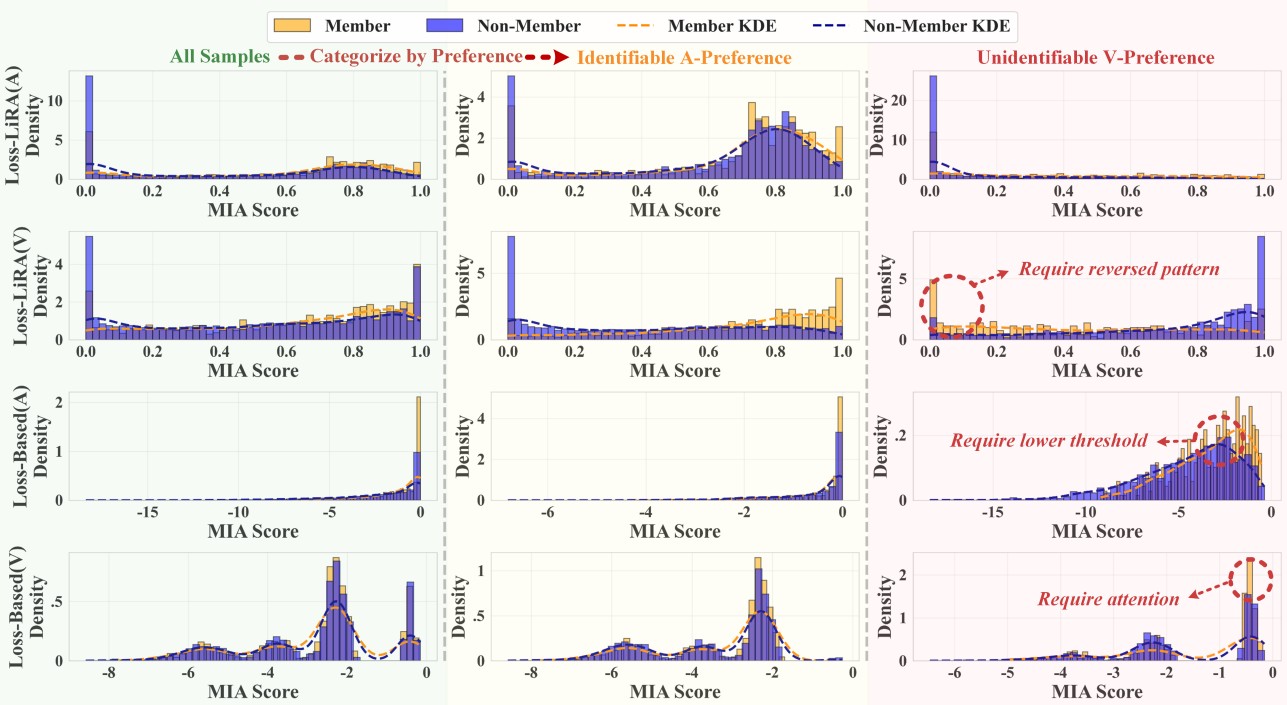

*Figure 1.* Density distribution and KDE curves of MIA scores for member and non-member samples from CREMA-D with distinct modality preferences. The rows correspond to the attack employed (Loss-Based and Loss-LiRA) for both audio and visual modalities, while columns categorize samples based on their modality preferences (A: Audio, V: Visual).

ples. Remarkably, when conducting loss-Based likelihood attacks on the visual modality, visually-preferred samples with lower scores are paradoxically more likely to be members, as the visual modality loss increases throughout the training process. These findings motivate us to design a modality-aware MIA method.

Although preference-based sample categorization improves membership discrimination, binary categorization is overly simplistic, as samples with identical preferences still exhibit varying cross-modal performance gaps. We therefore train an attack model using these gaps as discriminative features and decompose our objective into two tasks: (1) selecting appropriate attack scores for each sample based on its specific gap, and (2) calibrating the selected scores.

### 4.3. Attack Framework

Building upon the design intuition in Section 4.2, we propose a method that leverages cross-modal performance discrepancies to enhance MIA performance for MFL.

**Attack Model Construction.** Our attack model consists of a weight allocation module and a joint scoring module. The weight allocation module utilizes cross-modal performance disparities to generate adaptive weights. These weights are assigned to the corresponding modality-specific attack scores, which are derived from distinct computational pa-

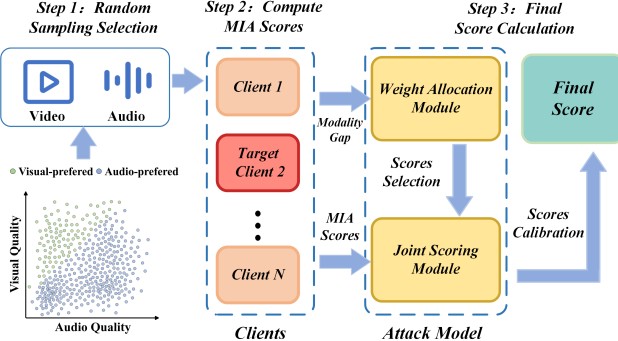

*Figure 2.* Our model performs adaptive MIA by selecting appropriate MIA scores and calibrating scores based on modality preferences of samples.

rameters tailored to each modality. Finally, the joint scoring module fuses these weighted scores to compute a final membership score, enabling the model to effectively distinguish between members and non-members by leveraging informative signals from each modality. We provide the detailed pipeline of our attack model in Figure 2.

**Attack Model Training.** The attacker first constructs a dataset for training the attack model. Specifically, the attacker leverages their own local training data to compute membership scores targeting different modalities using var-

ious MIA methods on their FL local model, constructing these scores as samples labeled as members. Concurrently, the attacker utilizes non-overlapping samples from other colluding clients to compute membership scores on their local model in MFL, constructing these as samples labeled as non-members. This collusion-based setup eliminates the need to eavesdrop on other clients' models for attack dataset construction, enabling immediate MIA deployment upon obtaining the target client's model. Moreover, this approach mitigates potential attack data contamination issues arising from sample overlap between the attacker and other clients.

Subsequently, the attacker trains the attack model using this dataset. Specifically, cross-modal performance gaps are fed into the weight allocation module to obtain weights for different MIA scores, and the weighted MIA scores are then passed to the joint scoring module to produce the final MIA score. Notably, we also treat the cross-modal performance gap itself as an MIA score, as we observe that samples with extreme performance gaps tend to be members.

**Performing Membership Inference.** Once the adversary intercepts the models uploaded by other participants, they compute initial MIA scores for a target sample by extracting model-specific parameters . These scores are then fed into the trained attack model to generate a final, calibrated membership score. To conclude the inference, the attacker selects an appropriate decision threshold according to False Positive Rate (FPR) and True Positive Rate (TPR) trade-off to determine whether the target sample is a member.

## 5. Experimental Evaluation

This section presents the empirical analysis of the proposed framework in terms of experimental setting, attack effectiveness, and robustness [2].

### 5.1. Experimental Setup

**Datasets & Models.** In this study, we employ three multimodal datasets: CREMAD (Cao et al., 2014), Balanced (Xia et al., 2023), and AVE (Tian et al., 2018), all of which consist of paired audio and video data. For each modality, we use ResNet-18 (He et al., 2016) for feature extraction, followed by a one-layer MLP for classification after fusion.

**Attack Model.** The weight allocation and joint scoring modules are built upon foundational MLP architectures. We explored different architectural and optimization choices, including two weight calculation mechanisms (Gating and Cross Attention (abbreviated as CrossAtt)) and two optimization strategies (Binary Cross-Entropy (BCE) and Asymmetric Loss (ASL)). In the following, we denote our method

---

[2]Code is available at https://github.com/chang2019RaceWithTime/MMIA

variants by combining these choices, e.g., BCE-Gating.

**MFL Setting.** We adopt the standard FedAvg algorithm (McMahan et al., 2017) and conduct experiments across various datasets, with the number of participants ranging from 3 to 20 clients. Furthermore, we consider a specialized non-IID scenario where clients exhibit varying cross-modal performance gaps.

**MIA Methods & Settings.** We implement a comprehensive range of MIA methods on individual modalities, including direct attacks based on Loss (Yeom et al., 2018), Grad-Cosine (Li et al., 2023), Grad-Norm (Nasr et al., 2018), and Grad-Diff (Li et al., 2023), as well as likelihood-based variants that exploit both Grad-Cosine and Loss metrics (Zhu et al., 2025) by quantifying the discrepancies of these values between the target participant and other clients. We perform attacks at training rounds exhibiting moderate overfitting, where training metrics steadily improve while test accuracy stagnates or marginally declines. While extreme overfitting typically yields higher MIA performance, we argue that such scenarios lack practical significance as models would never be trained to such extremes in practice.

**Evaluation Metrics.** We employ Balanced Accuracy and the Area Under the ROC Curve (AUC, where higher values indicate stronger attacks) to evaluate the overall effectiveness of the MIAs. Furthermore, we report the TPR at low FPR to reflect the attack's practical utility in realistic settings, where minimizing false alarms is critical.

For a comprehensive description of the experimental configuration, please refer to Appendix B.

### 5.2. Main Results

We first designate half of the clients as colluders and assume that all participant models are visible to evaluate the performance of our method. The comparison results of all attacks are presented in Tab. 3. We find that our method consistently outperforms alternative MIA baselines across all metrics, even though the peak performance for individual metrics is achieved by different methods. This superiority holds regardless of the strategy we employ and remains consistent across different MIA methods targeting any modality, eliminating the need to rely on specific attacks or target particular modalities. With the trained attack model, we can directly obtain the final MIA scores by aggregating all individual scores. To verify whether the attack model achieves our design goals of selecting appropriate MIA scores and calibrated scores for samples with different modality preferences, we extracted the outputs of the Gating network for various inputs and analyzed how different MIA scores and modality performance gaps influence the attack model's output. Fig. 3a illustrates that the attack model assigns different weights to various MIA scores based on the modality per-

*Table 3.* Complete performance comparison of MIA methods across various multimodal datasets (CRE: CREMAD, Bal: balance). The suffixes (e.g., -6, -10, -20) denote the number of clients in the FL system. Blue and red bold values denote the best baseline and the best proposed method in each column, respectively. **For each baseline, the three rows represent the attack results on multimodal fusion, audio, and visual modalities, respectively.**

| Attack Method | TPR @ 0.1% FPR | | | | TPR @ 1% FPR | | | | AUC | | | | Balanced Acc. | | | |
|---|---|---|---|---|---|---|---|---|---|---|---|---|---|---|---|---|
| | CRE-6 | CRE-10 | Bal-20 | AVE-6 | CRE-6 | CRE-10 | Bal-20 | AVE-6 | CRE-6 | CRE-10 | Bal-20 | AVE-6 | CRE-6 | CRE-10 | Bal-20 | AVE-6 |
| Cosine-LiRA | 0.00 | **4.33** | **4.32** | 1.30 | 10.39 | 10.01 | 10.76 | 7.49 | .7500 | **.7554** | .7200 | .6598 | 69.04 | 69.73 | **66.76** | 62.41 |
| | 1.88 | 2.39 | 3.51 | **2.28** | 11.47 | 7.92 | 9.37 | 4.72 | **.7551** | .7534 | .7217 | **.6731** | **69.49** | 70.18 | 66.65 | **64.36** |
| | 0.00 | 0.15 | 0.00 | 0.00 | 0.99 | 1.49 | 1.32 | 0.81 | .5140 | .5077 | .5567 | .5355 | 51.94 | 52.13 | 54.79 | 53.80 |
| Grad-Cosine | 4.75 | 3.89 | 3.51 | 1.63 | 12.28 | **13.45** | 10.47 | 6.68 | .7332 | .7484 | .7163 | .6472 | 67.30 | 68.03 | 66.34 | 60.98 |
| | **5.11** | 3.44 | **4.32** | 1.63 | **12.90** | 12.56 | **12.01** | **9.61** | .7366 | .7508 | **.7267** | .6570 | 67.45 | 68.60 | 66.63 | 62.70 |
| | 0.00 | 0.15 | 0.15 | 1.14 | 0.90 | 0.75 | 1.90 | 2.44 | .5185 | .5104 | .5577 | .5292 | 51.86 | 51.53 | 54.69 | 53.16 |
| Loss-LiRA | 0.90 | 1.64 | 1.76 | 0.33 | 6.63 | 4.63 | 6.44 | 2.93 | .6989 | .6685 | .6863 | .6481 | 65.40 | 62.77 | 63.67 | 61.02 |
| | 0.81 | 1.64 | 1.61 | 0.16 | 4.75 | 4.93 | 6.52 | 3.58 | .6787 | .6648 | .6863 | .6642 | 64.22 | 62.80 | 63.66 | 62.73 |
| | 0.00 | 0.00 | 0.15 | 0.16 | 0.99 | 0.45 | 1.32 | 1.30 | .5126 | .5025 | .5411 | .5197 | 51.64 | 51.57 | 53.21 | 53.25 |
| Loss-Based | 0.36 | 0.45 | 0.22 | 0.00 | 2.06 | 2.24 | 0.88 | 0.98 | .6629 | .6442 | .6116 | .6115 | 63.27 | 61.02 | 59.26 | 59.79 |
| | 0.09 | 0.90 | 0.07 | 0.00 | 2.15 | 1.94 | 0.95 | 0.81 | .6181 | .6227 | .5991 | .5878 | 60.16 | 59.40 | 57.87 | 58.00 |
| | 0.09 | 0.00 | 0.00 | 0.98 | 0.72 | 1.05 | 1.10 | 1.95 | .5003 | .5071 | .5219 | .5037 | 51.37 | 51.74 | 52.16 | 51.16 |
| Grad-Norm | 0.27 | 0.30 | 0.15 | 0.00 | 1.16 | 1.05 | 0.73 | 0.98 | .5414 | .5170 | .5046 | .5156 | 54.47 | 51.95 | 51.02 | 52.50 |
| | 0.27 | 0.30 | 0.15 | 0.00 | 1.34 | 1.05 | 0.95 | 0.98 | .5413 | .5170 | .5043 | .5169 | 54.41 | 51.96 | 51.05 | 52.43 |
| | 0.27 | 0.15 | 0.15 | 0.00 | 1.25 | 1.05 | 0.95 | 1.14 | .5405 | .5207 | .5070 | .5067 | 54.08 | 52.83 | 51.75 | 51.90 |
| Grad-Diff | 0.27 | 0.45 | 0.15 | 0.00 | 1.25 | 1.05 | 0.73 | 0.98 | .5417 | .5171 | .5045 | .5158 | 54.47 | 51.95 | 51.01 | 52.50 |
| | 0.27 | 0.45 | 0.15 | 0.16 | 1.25 | 1.05 | 0.95 | 1.14 | .5416 | .5171 | .5042 | .5170 | 54.41 | 51.96 | 51.04 | 52.43 |
| | 0.09 | 0.15 | 0.15 | 0.00 | 0.81 | 1.05 | 0.95 | 1.14 | .5395 | .5207 | .5069 | .5067 | 54.08 | 52.79 | 51.73 | 51.87 |
| **ASL-Gating** | 6.45 | **7.47** | **6.52** | 3.91 | 14.07 | 14.05 | 14.86 | 9.45 | .7802 | .7761 | .7524 | **.7242** | 70.69 | 69.91 | 68.02 | **65.77** |
| **BCE-Gating** | 6.27 | 7.17 | 6.30 | 3.26 | 15.32 | 15.70 | 14.71 | 9.93 | .7798 | **.7768** | .7538 | .7183 | 70.45 | **70.46** | **68.59** | 65.31 |
| **ASL-CrossAtt** | 6.54 | 7.17 | 6.30 | **4.89** | 15.95 | 14.20 | **15.23** | 9.77 | **.7804** | .7736 | **.7544** | .7187 | **71.04** | 70.13 | 68.09 | 65.58 |
| **BCE-CrossAtt** | **7.08** | **7.47** | 6.22 | 4.23 | **16.67** | **16.59** | **15.23** | **10.42** | .7788 | .7755 | .7516 | .7165 | 70.76 | 70.30 | 68.19 | 65.02 |

formance gaps of the samples. Taking the loss-Based MIA score for audio as an example, Fig. 3b shows that the final membership scores vary with the modality performance gap. Specifically, visual-preference samples have lower membership thresholds, while audio-preference samples more readily achieve higher confidence scores, which aligns with the observations in Figure 1. Both findings demonstrate that the attack model achieves our design goals.

We also verified that our method improves the recognition rate for samples with non-dominant modality preferences. Due to space constraints, these experimental results are provided in Appendix C.1. Furthermore, we conducted attacks based on auxiliary datasets in Appendix C.2, where the attacker cannot participate in MFL, which reveals unavoidable limitations compared to collusion attacks in MFL. Specifically, even assuming identical data distributions, models trained on auxiliary datasets rarely match the target's training state, fundamentally limiting attack effectiveness.

## 5.3. Robustness

This section demonstrates the robustness of our method across varying numbers of attacking clients, reference clients, degrees of overfitting, and non-IID scenarios.

**Number of Colluding Clients.** Under the balance-20 setting, we gradually reduce the number of colluding clients. Our results in Table 4 show that decreasing the number of

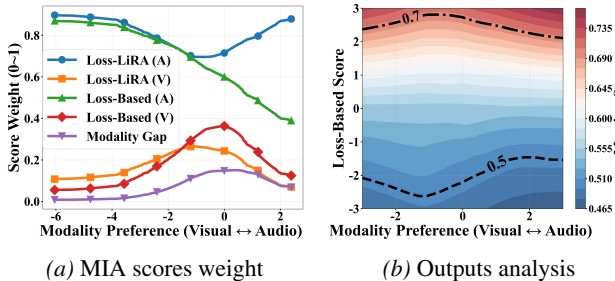

*(a)* MIA scores weight      *(b)* Outputs analysis

*Figure 3.* To more clearly demonstrate the internal mechanisms of the attack model, we employ five scores as inputs, including the modality performance gap and the attack methods shown in Figure 1, where (A) and (V) denote attacks targeting audio and visual modalities, respectively. (a) illustrates how the Gating network assigns weights to different MIA scores based on modality performance gap; (b) shows the influence of modality performance gaps on the final model output.

colluding clients only leads to a slight performance drop for our method, without affecting its superiority over baseline methods. This is because the constructed training dataset for the attack model is more than sufficient for training the low-dimensional attack model.

**Number of Reference Clients.** Likelihood-based attacks require models from other non-target clients as reference models. However, attackers cannot always obtain models from clients beyond the colluding clients. Under the

*Table 4.* Performance comparison with varying colluding clients (10 to 2) on Balanced dataset.

| Method | TPR @ 1% FPR | | | | | AUC | | | | |
|---|---|---|---|---|---|---|---|---|---|---|
| | 10 | 8 | 6 | 4 | 2 | 10 | 8 | 6 | 4 | 2 |
| ASL-Gating | 14.86 | 14.86 | 14.06 | 14.20 | 11.86 | .7524 | .7529 | .7510 | .7502 | .7466 |
| BCE-Gating | 14.71 | 14.93 | 14.49 | 14.35 | 12.96 | .7538 | .7530 | .7510 | .7504 | .7469 |
| ASL-CrossAtt | 15.23 | 15.08 | 13.91 | 14.64 | 12.01 | .7544 | .7537 | .7505 | .7492 | .7426 |
| BCE-CrossAtt | 15.23 | 14.49 | 14.71 | 15.01 | 14.20 | .7516 | .7509 | .7493 | .7507 | .7457 |

*Table 5.* Performance comparison with a varying number of reference models (6 to 0), excluding the models of the target and colluding clients. The suffixes (M), (A), and (V) denote attacks targeting the multimodal fusion, audio , and visual modality, respectively. Average denotes the mean performance of our four proposed methods.

| Attack Method | TPR @ 1% FPR | | | | AUC | | | |
|---|---|---|---|---|---|---|---|---|
| | 6 | 4 | 2 | 0 | 6 | 4 | 2 | 0 |
| Cosine-LiRA (M) | **10.01** | **8.07** | **5.98** | 0.00 | **.7554** | .7386 | .7249 | .7027 |
| Cosine-LiRA (A) | 7.92 | 6.88 | 5.38 | 0.00 | .7534 | **.7390** | **.7250** | **.7066** |
| Loss-LiRA (M) | 4.63 | 3.74 | 2.84 | **2.84** | .6685 | .6571 | .6470 | .6303 |
| Loss-LiRA (A) | 4.93 | 3.74 | 2.69 | **2.84** | .6648 | .6558 | .6476 | .6325 |
| **Average (Ours)** | **16.33** | **15.77** | **16.52** | **15.32** | **.7743** | **.7669** | **.7581** | **.7555** |

*Table 6.* MIA performance comparison across different overfitting degrees (Sli.: Slight, Mod.: Moderate, Sev.: Severe).

| Method | TPR @ 1% FPR | | | AUC | | |
|---|---|---|---|---|---|---|
| | Sli. | Mod. | Sev. | Sli. | Mod. | Sev. |
| Cosine-LiRA (M) | 2.69 | 10.01 | 16.29 | .6262 | **.7554** | .7999 |
| Cosine-LiRA (A) | 3.29 | 7.92 | 16.74 | .6390 | .7534 | **.8022** |
| Grad-Cosine (M) | 7.03 | **13.45** | 20.03 | .6305 | .7484 | .7970 |
| Grad-Cosine (A) | **8.37** | 12.56 | 20.03 | **.6492** | .7508 | .7967 |
| **Average (Ours)** | **7.70** | **15.14** | **22.01** | **.6473** | **.7755** | **.8576** |

CREMAD-10 setting with 3 colluding clients, we compare MIA performance across different numbers of reference models. As shown in Table 5, the performance of likelihood-based MIA methods degrades significantly as the number of reference models decreases. In contrast, our method remains largely unaffected due to its ability to adaptively select MIA scores, maintaining stable performance.

**Degree of Overfitting.** Under the CREMAD-10 setting, we evaluate MIA across training rounds with different degrees of overfitting. We examine three overfitting levels: slight (training accuracy begins to exceed test accuracy while test accuracy continues to rise), moderate (training metrics steadily improve while test accuracy stagnates or marginally declines), and severe (training loss/accuracy approaches 0/100% with significant test accuracy degradation). Our results in Table 6 show that our method achieves greater improvements over baselines in scenarios with more severe overfitting, where information leakage is more pronounced. Due to space constraints, we present only the top-performing baseline methods.

**Modality-Quality-Heterogeneous Non-IID Scenario.** Under the CREMAD-6 setting, we construct a non-IID scenario where clients with larger indices primarily hold samples with dominant modality preferences, while clients with smaller indices predominantly possess samples with non-dominant modality preferences. As shown in Table 7, we find that MIAs become easier in non-IID scenarios because each client's data distribution is no longer uniform. Our method consistently achieves superior performance over baselines across attacks on different clients, and even in

rare cases where it does not surpass baselines, the performance gap remains negligible. Furthermore, we observe varying MIA performance across different clients. Since samples with non-dominant modality preferences constitute a minority, MIAs against client 0 (who primarily holds non-dominant preference samples) achieve the highest success rate. Notably, Grad-Diff and Grad-Norm attacks, which were previously ineffective in our experiments, demonstrate substantial improvements in this scenario. This is because different clients have samples with divergent modality preferences, which diverge their modality optimization directions and thus enhance the discriminability of modality-specific gradients across clients.

**5.4. Defense Methods**

To investigate how to mitigate the vulnerability of MFL to MIAs and verify whether it stems from modality imbalance, we incorporate client-level DP into MFL, along with commonly used modality-balancing methods, including uni-modal loss regularization (Huang et al., 2025) and OGM (Peng et al., 2022), which adaptively balances the learning rates of different modalities.

As shown in Table 8, both DP and these modality-balancing methods can alleviate the vulnerability of MFL to MIAs, even though modality-balancing methods were originally designed to improve multimodal model performance rather than to protect privacy. Moreover, compared with DP, whose stronger privacy protection typically leads to degraded model utility, modality-balancing methods can even improve model performance. Therefore, when protecting privacy in MFL, modality-balancing methods should first be applied to suppress the additional privacy leakage caused by modality imbalance, followed by conventional privacy-preserving mechanisms such as DP.

This result further supports our claim that the vulnerability of MFL to MIAs originates from modality imbalance, since mitigating modality imbalance consistently reduces privacy leakage. We further validate this claim from the opposite direction. Specifically, we categorize multimodal datasets into two groups according to the quality gap across modalities,

*Table 7.* MIA performance comparison across target clients (indices 0, 2, 5) under modality quality heterogeneity.

| Method | TPR @ 1% FPR | | | AUC | | |
|---|---|---|---|---|---|---|
| | 0 | 2 | 5 | 0 | 2 | 5 |
| Cosine-LiRA (M) | **18.46** | 7.62 | 4.12 | .8769 | .7447 | **.7529** |
| Cosine-LiRA (A) | 15.68 | **8.33** | 4.74 | .8463 | .7101 | .7430 |
| Grad-Cosine (M) | 11.38 | 5.02 | **5.28** | .8429 | .7411 | .7390 |
| Grad-Cosine (A) | 11.38 | 7.62 | 5.19 | .8235 | .6994 | .7266 |
| Loss-LiRA (M) | 14.34 | 3.49 | 2.95 | **.9036** | .6723 | .7318 |
| Loss-LiRA (A) | 17.11 | 1.52 | 4.03 | .8971 | .6379 | .7304 |
| Loss-Based (M) | 2.69 | 6.36 | 2.60 | .8370 | **.7919** | .7397 |
| Loss-Based (A) | 0.63 | 2.60 | 1.43 | .7986 | .7375 | .6326 |
| Grad-Diff (M) | 0.09 | 2.69 | 0.00 | .7036 | .6737 | .5551 |
| Grad-Norm (M) | 0.09 | 2.69 | 0.00 | .7039 | .6735 | .5553 |
| Grad-Diff (V) | 3.14 | 2.01 | 2.24 | .6686 | .6603 | .6086 |
| Grad-Norm (V) | 3.14 | 2.51 | 2.33 | .6698 | .6606 | .6097 |
| **Average (Ours)** | **20.16** | **13.11** | **8.01** | **.8887** | **.7906** | **.7814** |

*Table 8.* Attack AUC and model performance under different defense methods, where $\sigma$ denotes the DP noise standard deviation.

| Category | Defense Method | AUC | Acc. |
|---|---|---|---|
| Modality Balancing | No Balancing | 0.7798 | 55.17 |
| | OGM | 0.5697 | 53.15 |
| | Uni-modal Loss | 0.6103 | 56.11 |
| Client-level DP | $\sigma = 0.1$ | 0.5254 | 34.63 |
| | $\sigma = 0.05$ | 0.5436 | 51.28 |
| | $\sigma = 0.01$ | 0.7819 | 54.63 |

namely datasets with relatively balanced modality quality and datasets with substantial modality-quality disparity. We find that multimodal models trained on datasets with larger modality-quality gaps are more prone to overfitting and are consequently more vulnerable to MIAs. Detailed results are provided in the appendix C.2.

## 6. Conclusion

While MFL achieves superior performance by leveraging complementary information across modalities, modality imbalance introduces a new attack surface that makes MFL more vulnerable to MIAs. However, existing MIA methods fail to fully exploit modality-specific information leaked in MFL, as they treat multimodal parameters as a whole without accounting for heterogeneous modality contributions. This work demonstrates that solely attacking the dominant modality achieves comparable performance with lower computational overhead, while attacks on other modalities broaden the attack perspective by revealing distinct membership patterns. Building upon these findings, we propose a nove modality-aware framework to identify member samples with diverse modality preferences by adaptively select-

ing attack modalities and calibrating inference results based on each sample's cross-modal performance discrepancies. Extensive experiments on three real-world datasets demonstrate that our method consistently outperforms baselines while maintaining robustness across various settings.

## Impact Statement

This paper presents work whose goal is to advance the field of Machine Learning. Our work exposes privacy vulnerabilities in multimodal federated learning systems, which helps raise awareness of these risks and ultimately motivates the development of stronger privacy protections.

## Acknowledgements

This work was supported in part by the Jiangsu Provincial Frontier Research and Development Program (No. BF2025067), the National Science and Technology Major Project of the Ministry of Science and Technology of China (No. 2026ZD1307100), the National Natural Science Foundation of China (No. 62471204, No. 62501150), the Natural Science Foundation of Jiangsu Province for Excellent Young Scientists (No. BK20250160), the Basic Research Program of Jiangsu (No. BK20251328), the Natural Science Foundation of the Jiangsu Higher Education Institutions of China (No. 24KJA510003), the Start-up Research Fund of Southeast University (No. RF1028625054), the Yongjiang Talent Program (No. 2024A-392-G), the Key Programs of Ningbo Municipal Natural Science Foundation (No. 2024J021), and the Key Project of the Zhejiang Provincial Natural Science Foundation (No. LZ26F020010).

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

# A. Proofs

## A.1. Proof of Theorem 3.1

Following the analytical approach of Zhang et al. (2023a); Cao et al. (2024), we further investigate the generalization error characteristics of multimodal models in this section. Let $(\mathbf{x}, y) \sim \mathcal{D}$ denote a multimodal sample drawn from the underlying data distribution $\mathcal{D}$. By the convexity of the loss function $L$, we have:

$$
\begin{aligned}
\mathbb{E}_{(\mathbf{x},y)\sim\mathcal{D}} L(f(\mathbf{x}), y) &= \mathbb{E}_{(\mathbf{x},y)\sim\mathcal{D}} L\left(\sum_{m=1}^{M} w^m f^m\left(\mathbf{x}^m\right), y\right) \\
&\leq \mathbb{E}_{(\mathbf{x},y)\sim\mathcal{D}} \sum_{m=1}^{M} w^m L\left(f^m\left(\mathbf{x}^m\right), y\right),
\end{aligned}
\tag{5}
$$

We can further decompose the right-hand side of the equation by covariance. With probability at least $1 - \delta$, we have

$$
\begin{aligned}
\mathbb{E}_{(\mathbf{x},y)\sim\mathcal{D}} L(f(\mathbf{x}), y) &\leq \sum_{m=1}^{M} \mathbb{E}_{(\mathbf{x},y)\sim\mathcal{D}}\left(w^m\right) \mathbb{E}_{(\mathbf{x},y)\sim\mathcal{D}}\left(L\left(f^m\left(\mathbf{x}^m\right), y\right)\right) + \mathrm{Cov}\left(w^m, L\left(f^m\left(\mathbf{x}^m\right), y\right)\right) \\
&\leq \sum_{m=1}^{M} \mathbb{E}_{(\mathbf{x},y)\sim\mathcal{D}}\left(w^m\right) \mathbb{E}_{(\mathbf{x},y)\sim\mathcal{D}}\left(\hat{L}\left(f^m\left(\mathbf{x}^m\right), y\right)\right) + \sum_{m=1}^{M} \mathbb{E}_{(\mathbf{x},y)\sim\mathcal{D}}\left(w^m\right) \mathfrak{R}_m(\mathcal{F}) \\
&\quad + \sum_{m=1}^{M} \mathrm{Cov}\left(w^m, L\left(f^m\left(\mathbf{x}^m\right), y\right)\right) + M\sqrt{\frac{\log(1/\delta)}{2N}},
\end{aligned}
\tag{6}
$$

where the first step utilizes the covariance identity $\mathbb{E}[ab] = \mathbb{E}[a]\mathbb{E}[b] + \mathrm{Cov}(a,b)$, allowing us to separate the interaction between the fusion weights $w^m$ and the modality losses.

## A.2. Proof of Theorem 3.2

We first assume that the loss function satisfies the $L$-smoothness condition, defined as follows:

**Assumption A.1.** The loss function $L(\cdot)$ is $L$-smooth with respect to the model parameters $\boldsymbol{\theta}$, i.e., for any $\boldsymbol{\theta}_1, \boldsymbol{\theta}_2$,

$$
\|\nabla L(\boldsymbol{\theta}_1) - \nabla L(\boldsymbol{\theta}_2)\| \leq L\|\boldsymbol{\theta}_1 - \boldsymbol{\theta}_2\|.
\tag{7}
$$

Let $\boldsymbol{\theta}_t^m$ denote the model parameters of modality $m$ at iteration $t$. We then obtain:

$$
\mathbb{E}\left[\hat{L}\left(\boldsymbol{\theta}_t^m\right)\right] \leq \mathbb{E}\left[\hat{L}(\boldsymbol{\theta}_{t-1}^m)\right] + \mathbb{E}\left[\left\langle\nabla\hat{L}(\boldsymbol{\theta}_{t-1}^m), \boldsymbol{\theta}_t^m - \boldsymbol{\theta}_{t-1}^m\right\rangle\right] + \frac{L}{2}\mathbb{E}\left[\left\|\boldsymbol{\theta}_t^m - \boldsymbol{\theta}_{t-1}^m\right\|^2\right].
\tag{8}
$$

The gradient update in late-fusion is:

$$
\boldsymbol{\theta}_t^m - \boldsymbol{\theta}_{t-1}^m = -\eta\frac{\partial\hat{L}\left(f(\mathbf{x}), y\right)}{\partial f(\mathbf{x})} w^m \frac{\partial f^m\left(\mathbf{x}^m\right)}{\partial\boldsymbol{\theta}_{t-1}^m},
\tag{9}
$$

where $f^m(\mathbf{x}^m)$ is the modality $m$ network output. To facilitate the convergence analysis without loss of generality, even when fusion involves high-dimensional vectors, we assume the fusion weight takes the form $\mathbf{w}^m = w^m \cdot \mathbf{1}$, where $w^m$ is a scalar and $\mathbf{1}$ is the all-ones vector. Under this assumption, the element-wise Hadamard product reduces to standard scalar multiplication. Accordingly, we define the virtual gradient as:

$$
\tilde{\nabla}\hat{L}(\boldsymbol{\theta}_t^m) := \frac{\partial\hat{L}(f(\mathbf{x}), y)}{\partial f(\mathbf{x})}\frac{\partial f^m(\mathbf{x}^m)}{\partial\boldsymbol{\theta}_t^m},
\tag{10}
$$

and the gradient update for unimodal training is:

$$
\nabla\hat{L}(\boldsymbol{\theta}_t^m) := \frac{\partial\hat{L}(f^m(\mathbf{x}^m), y)}{\partial f^m(\mathbf{x}^m)}\frac{\partial f^m(\mathbf{x}^m)}{\partial\boldsymbol{\theta}_t^m}.
\tag{11}
$$

We assume the gradient discrepancy is bounded for all $t$: $\mathbb{E}\left[\left\|\nabla\hat{L}(\boldsymbol{\theta}_t^m) - \tilde{\nabla}\hat{L}(\boldsymbol{\theta}_t^m)\right\|^2\right] \leq \lambda^m$, where $\lambda^m$ is a non-negative constant that captures the divergence between unimodal and multimodal training for modality $m$. The difference is determined by the discrepancy between $f^m(\mathbf{x}^m)$ and $f(\mathbf{x})$. Since $f(\mathbf{x})$ is a weighted combination of the modality outputs $f^m(\mathbf{x}^m)$, it is more similar to the outputs of modalities with larger weights. Consequently, the value of $\lambda^m$ decreases as $w^m$ increases. In particular, when $w^m = 1$, we have $\lambda^m = 0$. The first-order term in (8) is bounded as:

$$
\begin{aligned}
\mathbb{E}\left\langle \nabla\hat{L}(\boldsymbol{\theta}_{t-1}^m), \boldsymbol{\theta}_t^m - \boldsymbol{\theta}_{t-1}^m \right\rangle &= -\eta w^m \mathbb{E}\left\langle \nabla\hat{L}(\boldsymbol{\theta}_{t-1}^m), \nabla\hat{L}(\boldsymbol{\theta}_{t-1}^m) + \left(\tilde{\nabla}L(\boldsymbol{\theta}_{t-1}^m) - \nabla\hat{L}(\boldsymbol{\theta}_{t-1}^m)\right) \right\rangle \\
&= -\eta w^m \mathbb{E}\left[\|\nabla\hat{L}(\boldsymbol{\theta}_{t-1}^m)\|^2\right] - \eta w^m \mathbb{E}\left\langle \nabla\hat{L}(\boldsymbol{\theta}_{t-1}^m), \tilde{\nabla}L(\boldsymbol{\theta}_{t-1}^m) - \nabla\hat{L}(\boldsymbol{\theta}_{t-1}^m) \right\rangle \\
&\leq -\eta w^m \mathbb{E}\left[\|\nabla\hat{L}(\boldsymbol{\theta}_{t-1}^m)\|^2\right] + \frac{\eta w^m}{2}\mathbb{E}\left[\|\nabla\hat{L}(\boldsymbol{\theta}_{t-1}^m)\|^2 + \|\tilde{\nabla}L(\boldsymbol{\theta}_{t-1}^m) - \nabla\hat{L}(\boldsymbol{\theta}_{t-1}^m)\|^2\right] \\
&\leq -\frac{\eta w^m}{2}\mathbb{E}\left[\|\nabla\hat{L}(\boldsymbol{\theta}_{t-1}^m)\|^2\right] + \frac{\eta w^m}{2}\lambda^m.
\end{aligned}
\tag{12}
$$

where the first inequality follows from the Cauchy-Schwarz variation $-\langle \mathbf{a}, \mathbf{b} \rangle \leq \frac{1}{2}(\|\mathbf{a}\|^2 + \|\mathbf{b}\|^2)$. For the second-order term, applying the property $\|\mathbf{a} + \mathbf{b}\|^2 \leq 2\|\mathbf{a}\|^2 + 2\|\mathbf{b}\|^2$, we obtain:

$$
\begin{aligned}
\mathbb{E}\left[\frac{L}{2}\|\boldsymbol{\theta}_t^m - \boldsymbol{\theta}_{t-1}^m\|^2\right] &= \frac{L\eta^2(w^m)^2}{2}\mathbb{E}\left[\|\tilde{\nabla}L(\boldsymbol{\theta}_{t-1}^m)\|^2\right] \\
&= \frac{L\eta^2(w^m)^2}{2}\mathbb{E}\left[\|\nabla\hat{L}(\boldsymbol{\theta}_{t-1}^m) + (\tilde{\nabla}L(\boldsymbol{\theta}_{t-1}^m) - \nabla\hat{L}(\boldsymbol{\theta}_{t-1}^m))\|^2\right] \\
&\leq L\eta^2(w^m)^2 \mathbb{E}\left[\|\nabla\hat{L}(\boldsymbol{\theta}_{t-1}^m)\|^2\right] + L\eta^2(w^m)^2\lambda^m
\end{aligned}
\tag{13}
$$

Combining equations (12) and (13) into (8), we have:

$$
\mathbb{E}[\hat{L}(\boldsymbol{\theta}_t^m)] \leq \mathbb{E}[\hat{L}(\boldsymbol{\theta}_{t-1}^m)] - \frac{\eta w^m}{2}(1 - 2L\eta w^m)\mathbb{E}\left[\|\nabla\hat{L}(\boldsymbol{\theta}_{t-1}^m)\|^2\right] + \frac{\eta w^m}{2}(1 + 2L\eta w^m)\lambda^m.
\tag{14}
$$

Telescoping this inequality over $t = 1, 2, \ldots, T$ yields:

$$
\mathbb{E}[\hat{L}(\boldsymbol{\theta}_T^m)] \leq \mathbb{E}[\hat{L}(\boldsymbol{\theta}_0^m)] - \frac{\eta w^m}{2}(1 - 2L\eta w^m)\sum_{t=1}^{T}\mathbb{E}\left[\|\nabla\hat{L}(\boldsymbol{\theta}_{t-1}^m)\|^2\right] + \frac{T\eta w^m}{2}(1 + 2L\eta w^m)\lambda^m.
\tag{15}
$$

This completes the proof.

## B. Detailed Experimental Settings

### B.1. Datasets

**CREMAD** (Cao et al., 2014) is a multimodal dataset for emotion recognition, containing 7,442 clips (2-3s) from 91 actors speaking several short words. It covers six basic emotions: anger, disgust, fear, happiness, neutrality, and sadness.

**Balanced** (Xia et al., 2023) dataset contains 30,357 clips across 30 categories, integrating data from VGG-Sound, Kinetics-400, and YouTube. By selecting samples that satisfy specific cross-modal performance criteria, the Balanced dataset achieves a uniform distribution of modality preferences throughout the entire dataset.

**AVE** (Tian et al., 2018) comprises 4,143 YouTube videos across 28 classes with synchronized audio-visual tracks and second-level annotations. To build our experimental benchmark, we extract frames and audio from event-specific segments to form a labeled multimodal classification dataset.

### B.2. Attack Model

For weight generation, we developed two distinct architectural approaches. The Gating mechanism maps the modality gap signal directly into a weighting vector to modulate feature importance based on the gap's magnitude. In contrast, the

CrossAtt mechanism treats the modality gap as a query to probe the MIA scores, adaptively determining weights based on the correlation between the gap signal and the MIA scores for more refined score recalibration.

Binary Cross-Entropy (BCE) serves as the fundamental objective to establish a stable training baseline by measuring the standard discrepancy between predictions and ground truth. In contrast, Asymmetric Loss (ASL) is specifically employed to address the stringent performance requirement for TPR at low FPR in membership inference. By applying a higher focusing intensity and an asymmetric clipping mechanism to easy non-member samples, ASL effectively suppresses false positives.

### B.3. MFL Non-IID Setting

We construct a non-IID scenario characterized by modality quality heterogeneity. Specifically, we quantify each modality's quality by training unimodal models on the respective modality data of each multimodal sample. The training difficulty encountered for each modality serves as a direct proxy for its quality. By calculating the performance disparity between different modalities of the same sample, we rank and allocate the data to clients sequentially to establish a heterogeneous non-IID scenario. These data-level biases subsequently force the local models to exhibit diverse levels of modality imbalance, allowing us to evaluate the robustness of our framework under non-uniform modality-specific performance.

### B.4. Baseline MIA Methodologies

**Loss-Based** (Yeom et al., 2018): This method operates on the intuition that a trained model typically yields significantly lower prediction error on its training samples. By monitoring the loss values, the adversary can identify members as those samples whose error falls below a certain confidence threshold.

**Grad-Norm** (Nasr et al., 2018): This method focuses on the intensity of model updates. Since optimization algorithms tend to converge on training data, the resulting gradient magnitudes for members are generally smaller and more stable, whereas non-members often trigger larger, more erratic gradients due to their novelty to the model.

**Grad-Cosine** (Li et al., 2023): This approach quantifies the directional congruency between the gradient of a specific sample and a reference update direction, such as the parameter update of the target client. A high degree of cosine similarity indicates that the sample's contribution is highly consistent with the model's current state, strongly suggesting its membership.

**Grad-Diff** (Li et al., 2023): This approach exploits the geometric property that excluding a member's gradient from the collective update results in a notable reduction in its overall magnitude. Conversely, for non-members, this operation typically increases the update's intensity due to their geometric independence from the training participants' subspace.

**LiRA-based Attacks (Loss-LiRA and Cosine-LiRA)** (Zhu et al., 2025): These methods identify members by quantifying the discrepancy between the target participant's loss and cosine similarity and the reference behavior of other clients. By benchmarking against this cross-client baseline, the attack filters out the noise of data heterogeneity to highlight the statistical distinctiveness of local members relative to the rest of the federated clients. Due to the setting of colluding participants, our methodology does not strictly necessitate the acquisition of models from other clients for reference purposes. Instead, we can utilize models from other colluding participants as references.

### B.5. Evaluation Metrics

**AUC:** This metric quantifies the attack's overall ability to distinguish between members and non-members. A higher AUC indicates that the attack more consistently assigns higher membership scores to training samples than to unseen ones, thereby reflecting a more pronounced degree of privacy vulnerability across various confidence levels.

**Balanced Accuracy:** To ensure a fair evaluation, especially when the candidate member and non-member sets are of unequal sizes, we utilize Balanced Accuracy. This metric calculates the average of the correct classification rates for both member and non-member classes, preventing the performance results from being biased toward the majority class.

**TPR at Low FPR:** We report the TPR at low FPR (e.g., 0.1%) to evaluate the attack's practical utility in high-confidence regimes. This is critical in realistic security scenarios where the adversary must strictly limit false alarms to ensure that identified instances are indeed true members of the training set.

*Table 9.* Count of identified member samples categorized by modality preferences at 1% and 0.1% FPR. Corresponding row groups represent MIAs targeting multimodal fusion, audio, and visual modalities, respectively. (A: Audio-Preference, V: Visual-Preference)

| Attack Method | CREMAD-6 | | | | CREMAD-10 | | | | Balance-20 | | | | AVE-6 | | | |
|---|---|---|---|---|---|---|---|---|---|---|---|---|---|---|---|---|
| | 1% FPR | | 0.1% FPR | | 1% FPR | | 0.1% FPR | | 1% FPR | | 0.1% FPR | | 1% FPR | | 0.1% FPR | |
| | A | V | A | V | A | V | A | V | A | V | A | V | A | V | A | V |
| Cosine-LiRA | 101 | 15 | 0 | 0 | 61 | 6 | **27** | 2 | 121 | 26 | 50 | **9** | 39 | 7 | 7 | 1 |
| | 106 | **22** | 16 | 5 | 47 | 6 | 14 | 2 | 101 | 27 | 41 | 7 | 27 | 2 | **12** | 2 |
| | 9 | 2 | 0 | 0 | 9 | 1 | 1 | 0 | 12 | 6 | 0 | 0 | 5 | 0 | 0 | 0 |
| Grad-Cosine | 123 | 14 | 48 | 5 | **80** | **10** | 23 | **3** | 117 | 26 | 41 | 7 | 35 | 6 | 9 | 1 |
| | **126** | 18 | **51** | **6** | 74 | 10 | 20 | 3 | **128** | **36** | **52** | 7 | **52** | 7 | 10 | 0 |
| | 8 | 2 | 0 | 0 | 4 | 1 | 1 | 0 | 16 | 10 | 1 | 1 | 9 | 6 | 3 | 4 |
| Loss-LiRA | 68 | 6 | 9 | 1 | 27 | 4 | 8 | 3 | 66 | 22 | 17 | 7 | 16 | 2 | 2 | 0 |
| | 49 | 4 | 9 | 0 | 30 | 3 | 8 | 3 | 64 | 25 | 16 | 6 | 15 | 7 | 1 | 0 |
| | 9 | 2 | 0 | 0 | 3 | 0 | 0 | 0 | 15 | 3 | 2 | 0 | 6 | 2 | 1 | 0 |
| Loss-Based | 23 | 0 | 4 | 0 | 15 | 0 | 3 | 0 | 11 | 1 | 3 | 0 | 6 | 0 | 0 | 0 |
| | 24 | 0 | 1 | 0 | 13 | 0 | 6 | 0 | 13 | 0 | 1 | 0 | 5 | 0 | 0 | 0 |
| | 2 | 6 | 0 | 6 | 2 | 5 | 0 | 0 | 0 | 15 | 0 | 0 | 1 | **11** | 0 | **6** |
| Grad-Diff | 14 | 0 | 3 | 0 | 7 | 0 | 3 | 0 | 8 | 2 | 1 | 1 | 6 | 0 | 0 | 0 |
| | 14 | 0 | 3 | 0 | 7 | 0 | 3 | 0 | 11 | 2 | 1 | 1 | 7 | 0 | 1 | 0 |
| | 9 | 0 | 1 | 0 | 7 | 0 | 1 | 0 | 5 | 8 | 0 | 2 | 7 | 0 | 0 | 0 |
| Grad-Norm | 13 | 0 | 3 | 0 | 7 | 0 | 2 | 0 | 8 | 2 | 1 | 1 | 6 | 0 | 0 | 0 |
| | 15 | 0 | 3 | 0 | 7 | 0 | 2 | 0 | 11 | 2 | 1 | 1 | 6 | 0 | 0 | 0 |
| | 14 | 0 | 3 | 0 | 7 | 0 | 1 | 0 | 5 | 8 | 0 | 2 | 7 | 0 | 0 | 0 |
| **ASL-Gating** | 137 | 20 | 63 | 9 | 82 | 12 | 40 | **10** | 167 | 36 | **77** | 12 | 49 | 9 | 23 | 1 |
| **BCE-Gating** | 146 | 25 | 60 | 10 | 95 | 11 | 42 | 6 | 166 | 35 | 73 | 13 | 51 | **10** | 19 | 1 |
| **ASL-CrossAtt** | 151 | 27 | 67 | 6 | 82 | 13 | 39 | 9 | **167** | 39 | 71 | 12 | 53 | 7 | **28** | 2 |
| **BCE-CrossAtt** | **156** | **30** | **69** | **10** | **97** | **14** | **43** | 7 | 164 | **44** | 69 | **16** | **55** | 9 | 22 | **4** |

## C. More experiment

### C.1. MIA Accuracy across Dominant and Non-dominant Modality Preference Samples

Table 9 demonstrates that our approach not only improves the identification rate of samples with dominant modality preferences but also successfully addresses the previously overlooked non-dominant ones. Notably, at a 1% FPR, our method exhibits significant improvement in detecting non-dominant modality preference members compared to all baselines.We further observe that our method does not achieve the highest identification count for non-dominant modality preference samples on the AVE dataset. This is primarily attributed to the scarcity of such samples within this specific dataset.

### C.2. MIAs Based on Auxiliary Datasets

We observe that several existing MIA approaches against centralized models assume the attacker has access to an auxiliary dataset for training the attack model (He et al., 2024; Shi et al., 2024). To evaluate our method under this setting, we separate the attacking client from the original FL system and train two independent MFL systems, one with the attacker and one with non-attacking clients, using identical MFL configurations. The attacker constructs a training dataset from their own MFL system to train the attack model and performs MIA against target clients in the other FL system. Table 10 demonstrates that our method still achieves optimal performance in most cases under this setting.

Notably, our method does not outperform the baseline on the CREMAD-6 setting. This is because even with identical training configurations, the actual training dynamics of the two MFL systems differ. As shown in Table 11, MIA performance significantly improves when the attack models are trained using rounds that, though differing in index, are more closely aligned with the target MFL system's training state. This reveals a fundamental limitation of auxiliary-dataset-based attack models. Even assuming identical data distributions, models trained on auxiliary datasets rarely align perfectly with the target client's model, making it difficult for the attack model to achieve effective MIA against the target. Moreover, target clients may employ early stopping to prevent overfitting, making it harder for attackers to predict the target's training state and further increasing MIA difficulty.

We also consider scenarios where the auxiliary dataset and the target FL system's dataset exhibit heterogeneous modality quality. Specifically, we rank samples in the CREMAD dataset by the quality gap between their modalities and partition them into two datasets: one serving as the auxiliary dataset and the other as the target MFL system's dataset. One dataset represents an extreme case containing only samples with dominant modality preference, while the other is more balanced

*Table 10.* Performance comparison of auxiliary dataset-based attack methods across different datasets. Each row group for baselines represents MIAs targeting multimodal fusion, audio, and visual modality in that order.

| Attack Method | TPR @ 0.1% FPR | | | | TPR @ 1% FPR | | | | AUC | | | | Balanced Acc. | | | |
|---|---|---|---|---|---|---|---|---|---|---|---|---|---|---|---|---|
| | CRE-6 | CRE-10 | Bal-20 | AVE-6 | CRE-6 | CRE-10 | Bal-20 | AVE-6 | CRE-6 | CRE-10 | Bal-20 | AVE-6 | CRE-6 | CRE-10 | Bal-20 | AVE-6 |
| Cosine-LiRA | 0.00 | 0.00 | 2.27 | 0.00 | 0.00 | 4.33 | 12.66 | 0.00 | .6647 | .6494 | .7659 | .6581 | 63.66 | 61.69 | 70.21 | 63.26 |
| | 0.00 | 0.00 | 2.71 | 0.00 | 0.00 | 4.63 | **15.01** | 0.00 | .6790 | **.6636** | **.7877** | .6604 | **64.46** | **62.60** | **72.45** | **63.35** |
| | 0.00 | 0.15 | 0.15 | 0.00 | 0.00 | 1.05 | 0.81 | 0.00 | .5215 | .5046 | .5588 | .5276 | 52.53 | 51.77 | 54.59 | 53.18 |
| Grad-Cosine | 1.52 | 0.60 | 2.42 | 2.12 | 4.84 | 4.63 | 11.35 | 3.91 | .6639 | .6398 | .7553 | .6641 | 61.98 | 60.57 | 69.59 | 62.09 |
| | **1.70** | **0.90** | **4.03** | **3.26** | **6.09** | **6.28** | 14.71 | **5.05** | **.6847** | .6583 | .7787 | **.6658** | 63.24 | 61.33 | 71.47 | 62.23 |
| | 0.09 | 0.00 | 0.15 | 0.00 | 0.99 | 0.30 | 1.98 | 1.95 | .5215 | .5068 | .5528 | .5405 | 52.24 | 51.16 | 54.10 | 54.26 |
| Loss-LiRA | 0.00 | 0.30 | 1.32 | 0.00 | 0.00 | 4.48 | 8.78 | 0.00 | .5691 | .5881 | .7294 | .6349 | 56.56 | 57.10 | 66.80 | 61.91 |
| | 0.00 | 0.15 | 0.95 | 0.00 | 0.00 | 2.54 | 8.86 | 0.00 | .5652 | .5705 | .7515 | .6560 | 56.44 | 56.95 | 69.25 | 62.45 |
| | 0.00 | 0.00 | 0.22 | 0.00 | 0.00 | 1.35 | 1.39 | 0.00 | .5031 | .5006 | .5460 | .5076 | 51.55 | 51.15 | 53.50 | 52.53 |
| Loss-Based | 0.18 | 0.00 | 0.15 | 0.00 | 1.70 | 1.49 | 1.02 | 0.81 | .5735 | .5735 | .6600 | .6191 | 56.25 | 57.27 | 63.35 | 61.90 |
| | 0.09 | 0.00 | 0.22 | 0.00 | 1.70 | 1.94 | 1.46 | 0.65 | .5577 | .5449 | .6387 | .6000 | 54.52 | 54.56 | 61.03 | 58.72 |
| | 0.18 | 0.15 | 0.15 | 0.00 | 0.90 | 1.05 | 1.39 | 1.63 | .5113 | .5167 | .5214 | .5820 | 51.55 | 52.69 | 52.33 | 57.43 |
| Grad-Diff | 0.00 | 0.30 | 0.15 | 0.00 | 1.70 | 1.94 | 0.88 | 1.47 | .5275 | .5296 | .5062 | .5515 | 52.41 | 52.40 | 51.23 | 56.84 |
| | 0.00 | 0.30 | 0.15 | 0.00 | 1.70 | 1.94 | 0.88 | 1.30 | .5275 | .5294 | .5069 | .5518 | 52.48 | 52.52 | 51.01 | 56.80 |
| | 0.00 | 0.15 | 0.00 | 0.33 | 1.08 | 1.79 | 0.44 | 2.44 | .5240 | .5328 | .5002 | .5463 | 52.09 | 53.04 | 50.78 | 56.29 |
| Grad-Norm | 0.00 | 0.30 | 0.15 | 0.00 | 1.61 | 1.94 | 0.95 | 0.65 | .5274 | .5295 | .5063 | .5508 | 52.41 | 52.40 | 51.24 | 56.84 |
| | 0.00 | 0.30 | 0.15 | 0.00 | 1.61 | 1.94 | 0.88 | 0.65 | .5274 | .5293 | .5070 | .5513 | 52.50 | 52.50 | 51.02 | 56.80 |
| | 0.00 | 0.30 | 0.07 | 0.00 | 0.99 | 1.79 | 1.02 | 0.98 | .5239 | .5329 | .5002 | .5444 | 52.07 | 53.14 | 50.78 | 56.29 |
| **ASL-Gating** | 1.16 | 2.69 | 7.39 | 3.26 | 6.27 | 7.17 | 19.62 | 9.45 | .6480 | .7022 | .8137 | .7286 | 61.11 | 64.47 | 72.64 | 66.19 |
| **BCE-Gating** | 1.43 | 2.39 | 8.42 | 4.23 | 7.08 | 9.27 | 18.52 | 8.96 | .6615 | .7061 | .8131 | .7345 | 61.18 | 65.52 | 72.80 | 66.86 |
| **ASL-CrossAtt** | 0.99 | 2.09 | 8.27 | 5.21 | 5.82 | 6.88 | 20.64 | 9.61 | .6464 | .6999 | .8139 | .7270 | 60.91 | 64.86 | 72.66 | 66.06 |
| **BCE-CrossAtt** | 0.90 | 2.99 | 9.59 | 5.21 | 4.48 | 9.57 | 21.45 | 8.63 | .6572 | .7039 | .8152 | .7265 | 60.51 | 64.71 | 72.81 | 66.35 |

*Table 11.* Performance Improvement via Training State Matching on CREMAD-6. Bef. and Aft. denote results before and after matching the training state, and $\Delta$ represents the absolute gain.

| Method | TPR @ 0.1% FPR | | | TPR @ 1% FPR | | | AUC | | | Bal. Acc. | | |
|---|---|---|---|---|---|---|---|---|---|---|---|---|
| | Bef. | Aft. | $\Delta$ | Bef. | Aft. | $\Delta$ | Bef. | Aft. | $\Delta$ | Bef. | Aft. | $\Delta$ |
| ASL-Gating | 1.16 | 2.42 | +1.26 | 6.27 | 8.15 | +1.88 | .6480 | .7012 | +.0532 | 61.11 | 63.78 | +2.67 |
| BCE-Gating | 1.43 | 3.32 | +1.89 | 7.08 | 7.62 | +0.54 | .6615 | .6894 | +.0279 | 61.18 | 62.77 | +1.59 |
| ASL-CrossAtt | 0.99 | 1.61 | +0.62 | 5.82 | 7.62 | +1.80 | .6464 | .6822 | +.0358 | 60.91 | 62.60 | +1.69 |
| BCE-CrossAtt | 0.90 | 2.78 | +1.88 | 4.48 | 7.08 | +2.60 | .6572 | .6783 | +.0211 | 60.51 | 61.93 | +1.42 |
| **Average** | 1.12 | 2.53 | +1.41 | 5.91 | 7.62 | +1.71 | .6533 | .6878 | +.0345 | 60.93 | 62.77 | +1.84 |

with a more uniform distribution of modality preferences.

We observe that global models trained on the extreme dataset are more prone to overfitting and thus more vulnerable to MIA in Table 12. In contrast, models trained on the balanced dataset exhibit greater robustness against MIA due to more balanced modality training suppresses overfitting. This finding further validates our hypothesis that the vulnerability of multimodal models to MIA stems from modality imbalance. For our method, when the attack model is trained on the balanced scenario, it still achieves performance improvements in the extreme scenario. Conversely, training on the extreme scenario and attacking the balanced scenario fails to surpass the baseline. This demonstrates that our method is more effective in scenarios with greater information leakage and is relatively robust to variations in training data quality.

## C.3. Attack Cost

The additional overhead of training the attack model is minimal. On a RTX 4090 under the CREMAD-6 setting, training on 11,160 constructed samples takes less than 10 seconds, and inference on 3,721 samples takes only 0.08 seconds, owing to the small attack model and low sample dimensionality. Furthermore, compared to shadow-model-based MIA approaches that require approximately 2.11 hours to train a shadow FL model asynchronously, our method incurs no such cost, as the attacker and target client belong to the same FL system and are trained synchronously.

*Table 12.* Cross-scenario attack performance between Balanced and Extreme training scenarios. For each baseline group, the three rows represent results on multimodal fusion, audio, and visual scores in that order.

| Attack Method | Balanced Scenario | | | | Extreme Scenario | | | |
|---|---|---|---|---|---|---|---|---|
| | TPR@0.1 | TPR@1 | AUC | B-Acc | TPR@0.1 | TPR@1 | AUC | B-Acc |
| Cosine-LiRA | 0.00 | 0.00 | .6002 | **59.09** | 0.00 | 0.00 | .7729 | **73.09** |
| | 0.00 | 0.00 | **.6007** | 58.63 | 0.00 | 0.00 | .7705 | 72.94 |
| | 0.00 | 0.00 | .5157 | 51.75 | 0.00 | 0.00 | .5035 | 51.81 |
| Grad-Cosine | 0.63 | 3.58 | .5701 | 56.32 | 5.38 | **16.58** | **.8014** | 72.66 |
| | **0.81** | **3.94** | **.6007** | 58.63 | **6.27** | 15.14 | .7947 | 72.05 |
| | 0.00 | 0.09 | .5021 | 50.91 | 0.18 | 0.63 | .5074 | 51.70 |
| Loss-LiRA | 0.00 | 0.00 | .5600 | 54.59 | 0.00 | 0.00 | .7752 | 73.04 |
| | 0.00 | 0.00 | .5503 | 54.66 | 0.00 | 0.00 | .7589 | 70.78 |
| | 0.00 | 0.00 | .5191 | 51.98 | 0.00 | 0.00 | .5101 | 51.50 |
| Loss-Based | 0.00 | 0.54 | .5485 | 54.61 | 0.18 | 1.34 | .7159 | 70.20 |
| | 0.00 | 0.54 | .5462 | 54.15 | 0.36 | 1.97 | .6515 | 62.48 |
| | 0.00 | 1.08 | .5180 | 52.28 | 0.09 | 0.90 | .5152 | 52.52 |
| Grad-Diff | 0.18 | 0.63 | .5173 | 52.74 | 0.27 | 1.34 | .5259 | 54.46 |
| | 0.09 | 0.72 | .5174 | 52.56 | 0.27 | 1.43 | .5259 | 54.46 |
| | 0.00 | 1.34 | .5120 | 51.93 | 0.09 | 0.72 | .5286 | 54.61 |
| Grad-Norm | 0.00 | 0.54 | .5171 | 52.76 | 0.18 | 1.25 | .5252 | 54.46 |
| | 0.00 | 0.45 | .5171 | 52.56 | 0.18 | 1.25 | .5251 | 54.46 |
| | 0.00 | 0.72 | .5119 | 51.89 | 0.00 | 1.16 | .5287 | 54.61 |
| **ASL-Gating** | 0.36 | 2.69 | .5829 | 55.86 | 5.20 | 17.65 | .8252 | 75.21 |
| **BCE-Gating** | 0.36 | **4.21** | **.5923** | **56.95** | 4.39 | **18.64** | **.8253** | **75.42** |
| **ASL-CrossAtt** | **0.54** | 3.58 | .5911 | 56.46 | **5.47** | 14.16 | .8146 | 74.51 |
| **BCE-CrossAtt** | 0.45 | 3.67 | .5904 | 56.86 | 5.20 | 15.32 | .8119 | 74.61 |

