# OpenReview forum: "The Hidden Risk: Membership Inference Attacks on Multimodal Federated Learning via Modality Imbalance"
_ICML.cc/2026/Conference — ICML 2026 regular_

### Official Review · Reviewer_AafZ · 2026-02-21

**Soundness:** 4
**Presentation:** 3
**Significance:** 3
**Originality:** 3
**Overall Recommendation:** 5
**Confidence:** 3

**Summary:**

The paper presents a systematic study on Membership Inference Attacks (MIAs) within Multimodal Federated Learning (MFL). The authors demonstrate that multimodal models are inherently more vulnerable to MIAs due to modality imbalance and that attacks targeting individual modalities reveal distinct membership patterns. To exploit this, they propose a modality-aware MIA framework that leverages cross-modal performance gaps to adaptively weight and calibrate attack scores from different modalities. Experiments across three multimodal datasets show that the proposed method outperforms existing baselines in terms of both attack accuracy and TPR at low FPR. Overall, this work offers an insightful exploration of the privacy leakage risks associated with multimodal data.

**Compliance With Llm Reviewing Policy:**

Affirmed.

**Final Justification:**

The authors have addressed my concerns, and I will maintain my score.

**Key Questions For Authors:**

The paper identifies modality imbalance as the primary cause of vulnerability to MIAs in multimodal models. Could existing modality-balancing methods therefore serve as effective defenses against such attacks? Furthermore, how would their effectiveness compare to established defense strategies?

**Limitations:**

Yes.

**Strengths And Weaknesses:**

Strengths:
1. The authors present the first systematic study of MIAs against MFL, demonstrating that modality imbalance renders MFL inherently more vulnerable to such attacks. Furthermore, it demonstrates that MIAs targeting different modalities offer distinct attack perspectives.
2. The proposed method is insightful and intuitive. It effectively exploits the complex relationships among membership scores of different modalities by constructing an attack model that adaptively interprets and leverages these scores.
3. The paper is well-structured, with a clear logical flow from observation to theoretical analysis and experimental validation. Notably, Figure 1 illustrates the differences in score distributions, clearly motivating the proposed framework.

Weaknesses:
1. The method relies on the assumption of multiple colluding clients, which may not hold in practice. The paper lacks a discussion on the feasibility of the attack when only a single malicious client is available.
2. Although the method improves MIA performance, the paper lacks an analysis of the additional overhead incurred by training the attack model.

---

> ### Author Rebuttal · Authors · 2026-03-30
>
> > **W1**
>
> Our method remains applicable with a single malicious client, albeit requiring two
> additional assumptions adopted in prior work [1]. Specifically, the attacker can obtain
> other clients' uploaded local models as reference models, and the attacker and reference
> clients hold non-overlapping data. Under this setting, the single malicious client
> constructs member samples from its own model and non-member samples from reference models
> to train the attack model. As shown in the table below, our method still outperforms all
> baselines on most metrics under the CREMAD-6 setting. When multiple colluding clients are
> available, our method requires no additional assumptions, enabling a more efficient and
> effective attack.
>
> | **Method**     | **AUC**    | **Balanced Acc** | **TPR@0.1%FPR** | **TPR@1%FPR** |
> |----------------|------------|------------------|-----------------|---------------|
> | Best Baseline  | 0.7553     | 69.32           | 6.36          | 12.72        |
> | **Average (Ours)** | **0.7680** | **69.67**       | **5.74**      | **13.76**    |
>
> ---
>
> > **W2**
>
> The additional overhead of training the attack model is minimal. On a RTX 4090 under the
> CREMAD-6 setting, training on 11,160 constructed samples takes less than 10 seconds, and
> inference on 3,721 samples takes only 0.08 seconds, owing to the small attack model and
> low sample dimensionality. Furthermore, compared to shadow-model-based MIA approaches
> that require approximately 2.11 hours to train a shadow FL model asynchronously, our
> method incurs no such cost, as the attacker and target client belong to the same FL system
> and are trained synchronously.
>
> ---
>
> > **Q1**
>
> As demonstrated in our response to Reviewer iNc1's W4, modality balancing methods can
> effectively defend MFL against MIAs. We further conduct experiments with varying levels
> of client-level DP as a comparison, with results shown in the table below. Both DP and
> modality balancing reduce MFL's vulnerability to MIA. Stronger DP noise (larger noise
> standard deviation, denoted as Std in the table below) provides better privacy protection
> but incurs significant model performance degradation. In contrast, modality balancing
> reduces vulnerability while maintaining or even improving model performance. Therefore,
> when privacy protection is required in MFL, we recommend first applying modality balancing
> optimization to mitigate the additional overfitting caused by modality imbalance, and then
> imposing DP to further safeguard client privacy.
>
> | **Noise Std** | **AUC** | **Test Accuracy** |
> |---------------|----------------|-------------------|
> | 0.1           | 0.5254         | 34.63            |
> | 0.05          | 0.5436         | 51.28            |
> | 0.01          | 0.7819         | 54.63            |
>
> [1] Zhu G, Li D, Gu H, et al. Fedmia: An effective membership inference attack
> exploiting "all for one" principle in federated learning[C]//Proceedings of the IEEE/CVF
> Conference on Computer Vision and Pattern Recognition. 2025: 20643-20653.

---

> > ### Author Rebuttal · Reviewer_AafZ · 2026-04-02
> >
> > I thank the authors for their response, which addresses my main concerns. I will keep my score provided that the authors supplement a discussion of these limitations in the appendix of the final version.

---

> > > ### Author Response · Authors · 2026-04-03
> > >
> > > Thank you for your valuable comments. We will add a discussion of the limitations in the appendix of the final version as suggested. We believe this will provide a more comprehensive understanding of the proposed method.

---

### Official Review · Reviewer_SzCG · 2026-03-09

**Soundness:** 3
**Presentation:** 3
**Significance:** 3
**Originality:** 4
**Overall Recommendation:** 5
**Confidence:** 4

**Summary:**

This paper identifies that modality imbalance in multimodal federated learning (MFL) renders such systems inherently more susceptible to membership inference attacks (MIAs). Grounded in both theoretical analysis and empirical observation, the authors propose a modality-aware attack framework that adaptively selects and weights modality-specific attack scores based on each sample's cross-modal performance discrepancy to infer membership. Extensive experiments across diverse settings demonstrate the effectiveness and robustness of the proposed approach.

**Compliance With Llm Reviewing Policy:**

Affirmed.

**Final Justification:**

After carefully reviewing the authors' rebuttal and re-evaluating the manuscript in light of their responses, I am pleased to confirm that my initial concerns have been fully and satisfactorily addressed. I now believe this work meets the standards for acceptance and will make a valuable contribution to the conference. I therefore strongly recommend accepting this paper.

**Key Questions For Authors:**

1.	The threat model assumes the existence of colluding participants within the MFL. Is this assumption overly strong compared to existing MIA methods that rely on auxiliary datasets or single client attack settings?

2.	The theoretical analysis is restricted to late fusion architectures, and all experimental datasets contain only two modalities. Are the results of this paper still valid under other fusion strategies such as early or intermediate fusion, or in settings involving more than two modalities?

3.	The authors conduct attacks at the moderate overfitting stage and argue that the extreme overfitting setting is of limited practical significance. Please explain why the moderate overfitting stage is considered a more meaningful evaluation setting than the other overfitting stages.

**Limitations:**

Yes.

**Strengths And Weaknesses:**

Strengths:

1.	This paper tackles an important yet underexplored problem of privacy leakage in MFL induced by modality imbalance. Through generalization bound analysis and modality-specific MIA experiments, the authors derive insights that reveal previously overlooked privacy vulnerabilities and shed light on their underlying causes.

2.	The empirical evaluation is comprehensive, spanning multiple datasets and varied settings including different numbers of colluding clients, reference models, overfitting degrees, and non-IID distributions. Multiple metrics including AUC, balanced accuracy, and TPR at low FPR thresholds provide a thorough assessment of the proposed method.

Weaknesses:

1.	While the performance gains of the proposed method are consistent across settings, the absolute improvements remain limited in certain configurations. A more explicit discussion of the conditions under which the proposed method yields the most significant gains and those under which its advantages are marginal would better characterize the applicable scope of the method.

2.	The authors do not include experiments on defense mechanisms. Although the authors systematically expose the privacy leakage risks introduced by modality imbalance in MFL, no empirical analysis is provided on whether existing defense techniques such as differential privacy can effectively mitigate the additional attack surface.

3.	A typographical error appears in the abstract, where "dominants" should be corrected to "dominates."

---

> ### Author Rebuttal · Authors · 2026-03-30
>
> > **W1**
>
> The magnitude of improvement is primarily governed by the degree of model overfitting.
> Under severe overfitting, the model exposes more private information, and our method
> better captures these leaked signals, yielding larger gains over baselines. Under slight
> overfitting, the improvement becomes marginal and occasionally falls slightly below the
> best baseline. However, our method still offers practical value in this regime. By
> aggregating multiple MIA scores, it reliably achieves competitive performance without
> requiring the attacker to manually select among numerous MIA strategies, which may
> otherwise lead to suboptimal choices.
>
> ---
>
> > **W2**
>
> We have supplemented our work with additional discussions and experiments on defense
> mechanisms. In our response to Reviewer iNc1's W3, we discuss the applicability of our
> method under secret sharing and model masking scenarios. In our response to Reviewer
> iNc1's W4, we demonstrate that modality balancing methods can reduce MFL's vulnerability
> to MIA. In our response to Reviewer AafZ's Q1, we empirically validate the effectiveness
> of DP in protecting client privacy, and discuss the differences between DP and modality
> balancing as defense mechanisms and their complementary application.
>
> ---
>
> > **W3**
>
> Thank you for identifying this typographical error. We will correct "dominants" to
> "dominates" in the abstract in the next version of the manuscript.
>
> ---
>
> > **Q1**
>
> Our colluding assumption is not overly strong. In fact, our method relaxes certain
> assumptions required by existing federated MIA works without colluding clients [1].
> Specifically, it no longer requires access to other clients' local models as reference
> models, and avoids label conflicts caused by data overlap between the attacker and
> reference clients. Notably, our method remains applicable under a single malicious client
> setting (see Reviewer AafZ's W1) and under the auxiliary dataset setting (see Appendix
> C.2), outperforming baselines in both cases. When multiple colluding clients are
> available, our method requires no additional assumptions, enabling a more efficient and
> effective attack.
>
> ---
>
> > **Q2**
>
> We believe the theoretical conclusion that modality imbalance exacerbates overfitting is
> universal, as modality imbalance is an inherent characteristic of multimodal learning
> independent of fusion strategy or the number of modalities. We chose the two-modality
> late fusion architecture for theoretical analysis as its structure enables a clear
> mathematical formulation of the multimodal gradient update process, intuitively
> demonstrating that gradient updates are biased toward the dominant modality.
>
> To verify generality, we conduct experiments on the three-modality CMU-MOSEI dataset
> under both early and late fusion. We compute the cosine similarity between each modality
> encoder's gradients under unimodal and multimodal loss, and find that the dominant
> modality consistently yields significantly higher similarity than non-dominant modalities,
> confirming that our conclusions hold across fusion strategies and modality counts. The
> table below reports epoch-averaged gradient cosine similarity (Sim) and unimodal loss per
> modality.
>
> | **Model**    | **Sim Audio** | **Sim Visual** | **Sim Text** | **Loss Audio** | **Loss Visual** | **Loss Text** |
> |--------------|---------------|---------------|--------------|----------------|----------------|---------------|
> | Late Fusion  | 0.524         | 0.511         | 0.981        | 1.904          | 1.932          | 0.953         |
> | Early Fusion | 0.559         | 0.630         | 0.779        | 1.857          | 1.843          | 1.026         |
>
> ---
>
> > **Q3**
>
> We explain this along the progression of training stages.
>
> **Slight overfitting.** Training accuracy begins to exceed test accuracy, but test
> accuracy is still increasing. Clients would typically continue training in pursuit of
> better model performance.
>
> **Moderate overfitting.** Training accuracy continues to rise while test accuracy
> plateaus or slightly declines. Clients are most likely to stop training and deploy the
> model at this stage, making it the most practically meaningful setting for evaluating
> attack performance.
>
> **Severe overfitting.** Training accuracy approaches 100% while test accuracy drops
> substantially. Although MIAs achieve higher performance at this stage, clients would
> terminate training upon observing consistent performance decline, making severe
> overfitting unlikely in practical deployment.
>
> [1] Zhu G, Li D, Gu H, et al. Fedmia: An effective membership inference attack
> exploiting "all for one" principle in federated learning[C]//Proceedings of the IEEE/CVF
> Conference on Computer Vision and Pattern Recognition. 2025: 20643-20653.

---

> > ### Author Rebuttal · Reviewer_SzCG · 2026-04-02
> >
> > Thank you for the detailed rebuttal. My main concerns have been satisfactorily addressed. I will keep my positive score.

---

> > > ### Author Response · Authors · 2026-04-03
> > >
> > > Thank you for your valuable comments. We are glad that our responses have addressed your concerns. We will also incorporate the valuable points from the discussion into the manuscript to improve the clarity and depth of the paper.

---

### Official Review · Reviewer_iNc1 · 2026-03-12

**Soundness:** 4
**Presentation:** 3
**Significance:** 3
**Originality:** 3
**Overall Recommendation:** 4
**Confidence:** 4

**Summary:**

The authors focus on understanding and strengthening membership inference attacks (MIAs) in multimodal federated learning (MFL) under modality imbalance. The paper investigates how modality heterogeneity affects privacy leakage and proposes a modality-aware attack framework that adaptively selects and calibrates modality-specific attack scores.

The authors provides extensive experiments across 3 benchmarks (CREMAD, Balanced, and AVE) under various federated settings, including non-IID and modality-quality-heterogeneous scenarios. The paper further provides theoretical analysis linking modality imbalance to increased overfitting and MIA vulnerability.

**Compliance With Llm Reviewing Policy:**

Affirmed.

**Final Justification:**

After reading the rebuttals and other reviewers' comments. I will keep my original positive score.

**Key Questions For Authors:**

Key Questions For Authors
1. The introduction would benefit from a clearer distinction between standard FL and multimodal FL (MFL), particularly in terms of architectural assumptions and modality imbalance.

2. “…FL remains vulnerable to membership inference attacks (MIAs) due to the exchange of gradients and model parameters between clients and the server, which is absent in centralized learning”
This statement is inaccurate, as membership inference attacks are not unique to federated learning. MIAs were originally demonstrated in centralized settings [1], and their existence does not depend on gradient exchange between distributed parties. The authors should revise this statement for correctness.
[1]. Shokri, R., Stronati, M., Song, C., & Shmatikov, V. Membership Inference Attacks Against Machine Learning Models. In 2017 IEEE Symposium on Security and Privacy (SP).

3. The attack assumes access to modality-specific model updates. Under secure aggregation or scenarios where individual client updates are masked, would modality-level MIA still remain feasible?

4. The paper argues that modality imbalance inherently increases MIA vulnerability. Could the authors isolate this effect experimentally by enforcing balanced modality optimization (e.g., equalized training speed or fixed fusion weights) to verify that the vulnerability persists solely due to imbalance rather than general overfitting?

**Limitations:**

Yes

**Strengths And Weaknesses:**

Strengths
1. Clear research question and motivation. The paper clearly identifies a gap in the literature: existing MIAs in federated learning do not explicitly consider modality imbalance in multimodal settings.
2. Theoretical + empirical alignment. Theoretical analysis on modality imbalance and overfitting is reasonably connected to empirical findings regarding dominant modality vulnerability.
3. Well-designed attack framework. The proposed modality-aware attack model (with weight allocation and joint scoring modules) is conceptually coherent and improves upon baseline MIA methods.
4. Comprehensive experimental evaluation. The paper includes robustness analysis across varying numbers of colluding clients, reference clients, degrees of overfitting, and non-IID settings.

Weaknesses
1. Conceptual clarity in Introduction between FL and MFL
2. Potentially misleading statement about centralized learning.
3. Threat model realism under secure aggregation
4. Causality between Modality Imbalance and Increased MIA Vulnerability
Please read my below feedback for more detail

---

> ### Author Rebuttal · Authors · 2026-03-30
>
> > **W1 / Q1**
>
> Thanks for the suggestion. We will add the following to the introduction to clarify this
> distinction:
>
> - **Architecture.** Unlike unimodal FL where clients share a single model's parameters,
>   each client in MFL maintains multiple modality-specific encoders and a classifier. Each
>   encoder extracts modality-specific features, which are fused and passed to the
>   classifier for prediction.
>
> - **Modality imbalance.** Existing MIAs overlook the multimodal setting, where parameters
>   from different encoders exhibit heterogeneous optimization speeds and performance, a
>   phenomenon known as modality imbalance.
>
> ---
>
> > **W2 / Q2**
>
> Thanks for pointing this out. We intended to convey that gradient exchange in FL
> introduces attack surfaces that do not exist in centralized learning. We will revise the
> statement to:
>
> > "Despite keeping raw data local, FL remains vulnerable to membership inference attacks
> > (MIAs), as the exchange of gradients and model parameters between clients and the server
> > exposes additional attack surfaces compared to centralized learning."
>
> ---
>
> > **W3 / Q3**
>
> Our attack remains feasible under both secret sharing, a representative secure
> aggregation method, and masked modality updates.
>
> **Secret sharing.** In secret sharing, clients protect their updates with mutually
> negotiated masks that cancel upon aggregation. Our attack remains viable in both cases:
> (1) When colluding clients share the masking scheme with the target client, they can
> losslessly recover the target's updates and proceed with MIA as usual. (2) When the
> target client's updates cannot be recovered, the attacker can still perform MIA on the
> global model to determine whether a sample was used by any client, though without
> client-level attribution.
>
> **Masked modality updates.** Under the CREMAD-6 setting, we evaluate our attack in the
> scenario where one modality is masked, performing MIA using only the remaining modality
> updates (results shown in the table below). Compared with Table 3 of the manuscript, we
> find that: (1) When the non-dominant modality is masked, performance drops only slightly,
> as the dominant modality still exposes sufficient membership information. (2) When the
> dominant modality is masked, our method remains competitive and substantially outperforms
> baselines under the same condition. We note this scenario is uncommon in practice, as
> clients are incentivized to upload their better-trained modality updates to improve global
> model performance [1].
>
> We additionally discuss performance under differential privacy in response to Reviewer
> AafZ's Q1.
>
> | **Remaining Modality** | **AUC** | **Balanced Acc** | **TPR@0.1%FPR** | **TPR@1%FPR** |
> |------------------------|---------|------------------|-----------------|---------------|
> | Audio                  | 0.7752    |   70.09       | 5.60        | 15.68       |
> | Visual                 | 0.6549    |   61.15       | 0.27        | 1.93       |
>
> ---
>
> > **W4 / Q4**
>
> We provide both forward and reverse evidence to verify that modality imbalance introduces
> additional overfitting, leading to the vulnerability of MFL to MIA.
>
> **Forward verification.** We split CREMAD into two subsets based on modality quality
> distribution: one with extreme imbalance and one with relatively balanced modality
> quality. As shown in Table 11 of our manuscript, models trained on the balanced subset
> are less susceptible to MIA, while those trained on the extremely imbalanced subset
> exhibit severe overfitting and are substantially more vulnerable to MIAs, confirming that
> imbalanced modality optimization drives MIA vulnerability.
>
> **Reverse verification.** We incorporate two modality balancing methods into MFL: OGM
> [2], which dynamically adjusts training speed based on inter-modal performance
> discrepancy, and uni-modal loss [3], which decouples fusion weights from uni-modal
> parameter updates. As shown in the table below, under comparable model test accuracy,
> both methods substantially reduce attack AUC, further confirming that the vulnerability
> stems from modality imbalance rather than general overfitting.
>
> | **Balancing Method** | **Attack AUC** | **Test Accuracy** |
> |----------------------|----------------|-------------------|
> | No Balancing         | 0.7798         | 55.17        |
> | OGM                  | 0.5697         | 53.15            |
> | Uni-modal Loss       | 0.6103         | 56.11       |
>
> [1] Yuan L, Han D J, Wang S, et al. Communication-efficient multimodal federated learning: Joint modality and client selection[J]. IEEE Transactions on Mobile Computing, 2026.
>
> [2] Peng X, Wei Y, Deng A, et al. Balanced multimodal learning via on-the-fly gradient
> modulation[C]//Proceedings of the IEEE/CVF conference on computer vision and pattern
> recognition. 2022: 8238-8247.
>
> [3] Huang C, Wei Y, Yang Z, et al. Adaptive unimodal regulation for balanced multimodal
> information acquisition[C]//Proceedings of the Computer Vision and Pattern Recognition
> Conference. 2025: 25854-25863.

---

> > ### Author Rebuttal · Reviewer_iNc1 · 2026-04-02
> >
> > The authors addressed my concerns, but some adjustment needs to be done based on the submission version. I will maintain my positive score.

---

> > > ### Author Response · Authors · 2026-04-03
> > >
> > > Thank you for your valuable comments. We will incorporate the suggested adjustments and add the relevant discussion in the next revised manuscript. We believe these revisions will make the paper more logically coherent and convincing.

---

### Decision · Program_Chairs · 2026-04-30

**Decision:**

Accept (regular)

**Comment:**

The paper received uniformly positive reviews, with two accepts and one weak accept, and the reviewers’ concerns were largely resolved after rebuttal. This paper presents a systematic study of membership inference attacks in multimodal federated learning, showing that modality imbalance increases privacy leakage and proposing a modality-aware attack framework that adaptively combines modality-specific attack signals. Reviewers appreciated the clear motivation, the strong alignment between theoretical analysis and empirical findings, the coherent and effective attack design, and the comprehensive evaluation across multiple datasets and federated settings. The work was viewed as an important and timely contribution that exposes a previously underexplored privacy risk in MFL and provides a technically sound attack framework for analyzing it.
The main concerns focused on clarifying the distinction between standard FL and multimodal FL, refining some claims in the introduction, discussing the realism of the collusion-based threat model and secure aggregation scenarios, and providing stronger analysis of defense mechanisms such as modality balancing and differential privacy. Reviewers also requested clearer discussion of the conditions under which the attack gains are most significant, the generality of the conclusions beyond late-fusion two-modality settings, and the overhead of training the attack model. These issues were effectively addressed in the rebuttal through added experiments, clarifications, and discussion, including analysis under balancing methods, DP, secure aggregation, single-malicious-client settings, and broader multimodal architectures. Overall, the paper makes a meaningful and well-supported contribution to privacy analysis in multimodal federated learning, and the AC recommends accepting the paper.